# circNDUFB2 inhibits non-small cell lung cancer progression via destabilizing IGF2BPs and activating anti-tumor immunity

Botai Li[1,5], Lili Zhu[1,5], Chunlai Lu[2,5], Cun Wang [3], Hui Wang[3], Haojie Jin[3], Xuhui Ma[3], Zhuoan Cheng[1], Chengtao Yu[1], Siying Wang[3], Qiaozhu Zuo[3], Yangyang Zhou[3], Jun Wang[3], Chen Yang[3], Yuanyuan Lv[3], Liyan Jiang[4 ✉] & Wenxin Qin [1,3 ✉]

Circular RNAs (circRNA) are a class of covalently closed single-stranded RNAs that have been implicated in cancer progression. Here we identify *circNDUFB2* to be downregulated in non-small cell lung cancer (NSCLC) tissues, and to negatively correlate with NSCLC malignant features. Elevated *circNDUFB2* inhibits growth and metastasis of NSCLC cells. Mechanistically, *circNDUFB2* functions as a scaffold to enhance the interaction between TRIM25 and IGF2BPs, a positive regulator of tumor progression and metastasis. This TRIM25/circNDUFB2/IGF2BPs ternary complex facilitates ubiquitination and degradation of IGF2BPs, with this effect enhanced by N[6]-methyladenosine (m[6]A) modification of *circNDUFB2*. Moreover, *circNDUFB2* is also recognized by RIG-I to activate *RIG-I-MAVS* signaling cascades and recruit immune cells into the tumor microenvironment (TME). Our data thus provide evidences that *circNDUFB2* participates in the degradation of IGF2BPs and activation of anti-tumor immunity during NSCLC progression via the modulation of both protein ubiquitination and degradation, as well as cellular immune responses.

[1] State Key Laboratory of Oncogenes and Related Genes, Shanghai Cancer Institute, Renji Hospital, Shanghai Jiao Tong University School of Biomedical Engineering, 200032 Shanghai, China. [2] Department of Thoracic Surgery, Zhongshan Hospital, Fudan University, 200032 Shanghai, China. [3] Shanghai Cancer Institute, Renji Hospital, Shanghai Jiao Tong University School of Medicine, 200032 Shanghai, China. [4] Department of Respiratory Medicine, Shanghai Chest Hospital, Shanghai Jiao Tong University, 200030 Shanghai, China. [5] These authors contributed equally: Botai Li, Lili Zhu, Chunlai Lu. ✉email: jiang_liyan2000@126.com; wxqin@sjtu.edu.cn

Lung cancer is one of the most commonly diagnosed cancer and the leading cause of cancer deaths in most countries[1]. Non-small cell lung cancer (NSCLC) represents approximately 85% of lung cancer cases, including lung adenocarcinoma (LUAD), lung squamous cell carcinoma (LUSC), and large cell carcinoma histologic subtypes[2,3]. Despite advances in treatment of NSCLC, overall cure and survival rate for the patients remain low[4]. It is necessary to explore and understand molecular mechanism of development and progression for NSCLC to improve the outcome of patients with NSCLC.

circular RNAs (circRNAs) comprise a class of regulatory RNAs with a covalently closed single-stranded loop conformation that are produced by precursor mRNA (pre-mRNA) backsplicing of exons[5–7]. Recent studies have shown that circRNAs are involved in multiple biological processes, such as autoimmune disease[8,9], brain development[10], and tumorigenesis[11]. The landscape and action of circRNAs have been widely reported in various types of cancer, suggesting circRNAs may be potential biomarkers or therapeutic targets[12–16]. In lung cancer, circPRKCI increases the level of E2F7 by competitively binding to both miR-545 and miR-589, and finally promotes tumorigenesis of LUAD[17]. circTP63 promotes cell proliferation of LUSC by competitively sponging to miR-873-3p and upregulating FOXM1 (ref. [18]). However, mechanism of circRNAs reported in these papers mainly focuses on the role of them by acting as miRNA sponges. Further insight into the role and mechanism of circRNAs for NSCLC may contribute to understand the development and progression of NSCLC.

Insulin-like growth factor 2 mRNA-binding proteins (IGF2BPs) belong to an evolutionarily conserved family of RNA-binding oncofetal proteins, including IGF2BP1, IGF2BP2, and IGF2BP3 in the human genome[19]. The similarities of these three IGF2BP proteins canonical structures confer them similar biochemical functions[20]. IGF2BPs are highly expressed in the embryonic tissues and with low expression in adult tissues, in contrast become increased or de novo synthesized in several malignant cancers[19]. IGF2BPs are upregulated in NSCLC and a high level of them predicts poor prognosis. Overexpression of them promotes tumor growth and metastasis[21]. Function and mechanism of IGF2BPs involved in tumor progression have been characterized; however, degradation mechanism of IGF2BP proteins is still unclear.

Recent study has shown that cia-cGAS binds cGAS to block the enzymatic activity of it, and protects dormant LT-HSCs from cGAS-mediated exhaustion[9]. A subset of circRNAs with double-stranded structure prefers to bind and suppress activation of protein kinase R (PKR), thereby inhibiting innate immune responses[8]. Transfection of certain artificial circRNAs into cells leads to innate immunity gene expression that confers protection against viral infection[22]. In vitro-generated circRNA inhibits OVA-expressing tumors growth through serving as vaccine adjuvant to induce antigen-specific T cell activation and antibody production[23]. However, behavior and mechanism of immune responses induced by circRNAs in NSCLC development and progression have not been reported.

In the present study, we identify a circRNA derived from NDUFB2, hsa_circ_0007518 designated as circNDUFB2, by analyzing expression profiles of circRNAs in NSCLC. circNDUFB2 is frequently downregulated in NSCLC compared with matched adjacent nontumorous tissues, and decrease of QKI contributes to the downregulation of circNDUFB2. Functionally, circNDUFB2 inhibits NSCLC progression through destabilizing IGF2BPs and activating anti-tumor immunity. Our data suggest that circNDUFB2 may exert as a potential tumor suppressor to involve in NSCLC progression.

## Results

**circNDUFB2 is downregulated in NSCLC.** We analyzed expression profiles of circRNAs in three paired samples of NSCLC by Arraystar Human Circular RNA Microarray which contains 5396 probes specific for human circRNAs backsplice junction region. A total of 109 dysregulated circRNAs were identified in NSCLC tissues (GEO accession number: GSE158695, https://www.ncbi.nlm.nih.gov/geo/query/acc.cgi?acc=GSE158695) (Supplementary Data 1), of which 33 circRNAs were upregulated and 76 circRNAs were downregulated (Fig. 1a). We confirmed the differential expression of the top five dysregulated circRNAs in other 52 paired samples of NSCLC by quantitative reverse transcription PCR (qRT-PCR). Results showed that hsa_circ_0007518 which derived from NDUFB2, designated as circNDUFB2, was the most significantly downregulated circRNA (with average fold change of −8.82, $P < 0.0001$) in these five candidate circRNAs (Fig. 1b and Supplementary Fig. 1). We next analyzed correlation between circNDUFB2 expression and clinicopathologic features in patients with NSCLC and found that high expression of circNDUFB2 was negatively associated with tumor size, lymph node metastasis, and stage in NSCLC patients (Supplementary Table 1). Our results indicate that circNDUFB2 is frequently downregulated in NSCLC, and it negatively correlates with NSCLC malignant features.

**Characteristics of circNDUFB2 in NSCLC cells.** circNDUFB2 is generated from the exons 2–3 of NDUFB2 gene with a length of 249 nt. The backsplice junction site of circNDUFB2 was amplified using divergent primers and confirmed by Sanger sequencing (Fig. 1c). The sequence is consistent with circBase database annotation (http://www.circbase.org/). PCR analysis showed that circNDUFB2 could be amplified by divergent primers in cDNA reverse-transcribed from random hexamers, but not oligo(dT)$_{18}$ primers (Fig. 1d). Additionally, no product could be amplified from genomic DNA (Fig. 1d). Resistance to digestion with RNase R exonuclease confirmed that circNDUFB2 harbors a closed loop structure (Fig. 1e). Treatment by Actinomycin D showed that circNDUFB2 was stable in comparison to NDUFB2 mRNA (Fig. 1f). Nuclear and cytoplasmic fractionation as well as fluorescence in situ hybridization (FISH) examination revealed that circNDUFB2 was predominantly localized in the cytoplasm (Fig. 1g, h). These results demonstrate that circNDUFB2 is a bona fide circRNA.

**QKI promotes the biogenesis of circNDUFB2 in NSCLC cells.** The biogenesis of circRNAs in cells can be regulated both co- and post-transcriptionally[5,7]. Both circNDUFB2 and NDUFB2 were derived from NDUFB2 pre-mRNA, while NDUFB2 mRNA was slightly upregulated in NSCLC (Supplementary Fig. 2a). Therefore, we speculated that downregulation of circNDUFB2 occurs post-transcriptionally in NSCLC. The biogenesis of exon-derived circRNAs can be modulated by RNA-binding proteins post-transcriptionally[13,24,25]. We hypothesized that the downregulation of circNDUFB2 may be resulted from the decrease of certain RNA-binding proteins which are involved in generating circRNAs. The RNA-binding protein quaking (QKI) promotes circRNAs biogenesis via binding to intronic QKI response elements (QRE) flanking circRNA-forming exons[25]. QRE is a bipartite consensus sequence NACUAAY-N$_{1−20}$-UAAY, which contains a core sequence NACUAAY (Y, pyrimidine) and a half-site UAAY separated by 1–20 nt[26,27]. We found QKI was significantly downregulated in NSCLC (Supplementary Fig. 2b). Furthermore, we searched for sequences that match potential QRE in the introns flanking the circNDUFB2-forming exons. Two putative QREs were located in upstream and downstream of the circNDUFB2-forming splice sites, respectively (Supplementary Fig. 2c). Next, we performed RNA immunoprecipitation (RIP)

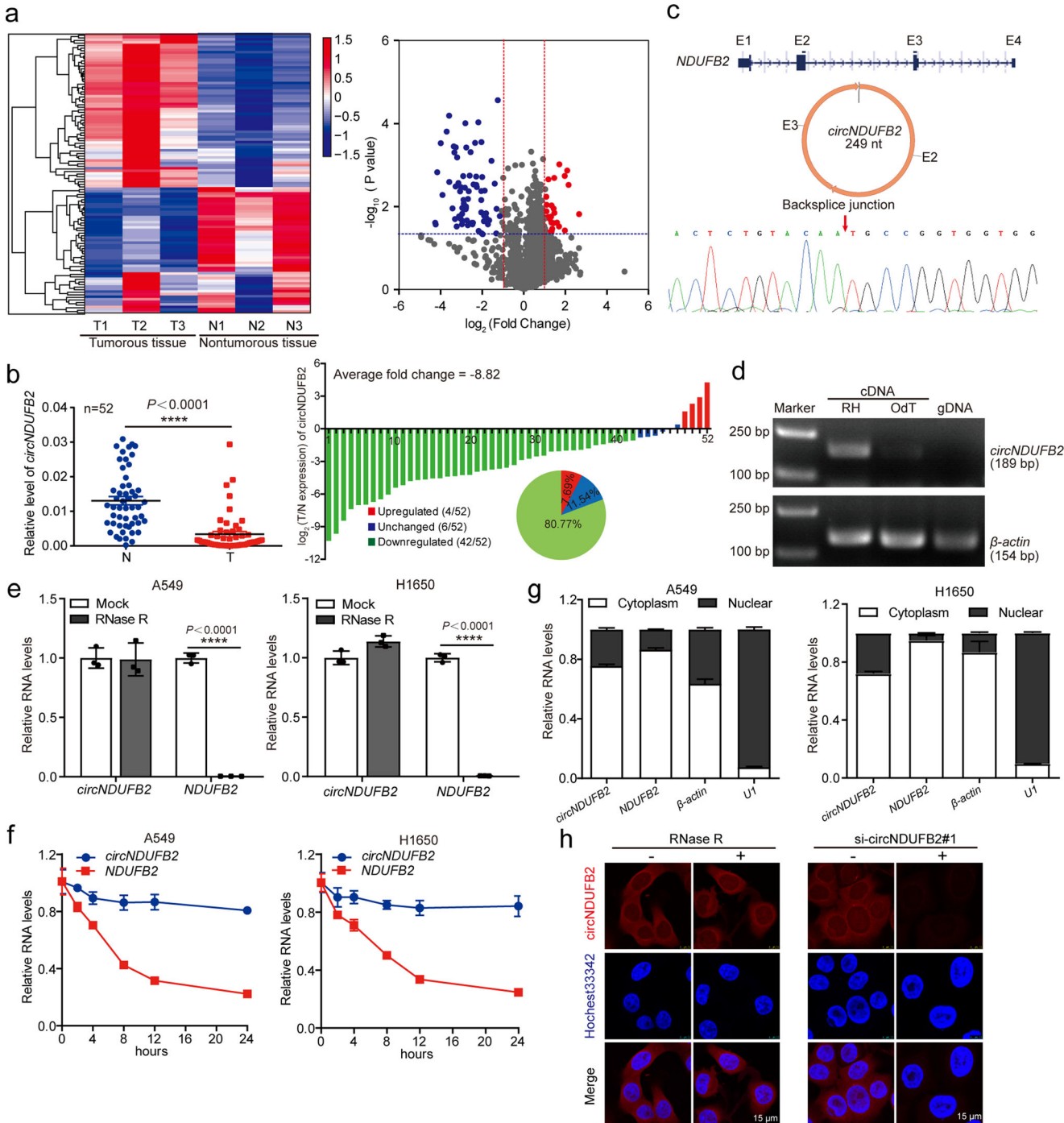

**Fig. 1 circRNA expression profiles in NSCLC and characterization of *circNDUFB2*. a** A heatmap and volcano plot show 109 differentially expressed circRNAs in three paired samples of NSCLC by Arraystar Human Circular RNA Microarray. Cut off is Log$_2$ (fold change) > 1 or <−1.0, $P$ value < 0.05. **b** Analysis for RNA levels of *circNDUFB2* in additional 52 paired samples of NSCLC (left). Expression proportions of *circNDUFB2* in histogram and pie chart (right). Log$_2$ (T/N expression) value > 1 as significantly higher expression, which <−1 as significantly lower expression, and between −1 and 1 as no significant change. N nontumorous tissue, T tumorous tissue. Data are presented as mean ± s.d. $P$ values are calculated by paired two-sided $t$-test. $n = 52$ biologically independent paired tissues of NSCLC. **c** The backsplice junction site of *circNDUFB2* was identified by Sanger sequencing. **d** PCR analysis for *circNDUFB2* and *β-actin* in cDNA and genomic DNA. RH random hexamers, OdT oligo(dT)$_{18}$ primers, gDNA genomic DNA. Two independent experiments were carried out with similar results. **e** Analysis for RNA levels of *circNDUFB2* and *NDUFB2* after treatment with RNase R. $n = 3$ biologically independent samples. **f** Analysis for RNA abundance of *circNDUFB2* and *NDUFB2* treated with Actinomycin D (2 μg/ml) at the indicated time point. $n = 3$ biologically independent samples. **g** RNA levels of *circNDUFB2*, *NDUFB2*, *β-actin*, and *U1* in the nuclear and cytoplasmic fractions of A549 and H1650 cells. $n = 3$ biologically independent samples. **h** RNA FISH analysis for *circNDUFB2* in A549 cells, scale bar = 15 μm. Two independent experiments were carried out with similar results. Data are presented as mean ± s.d in **e**–**g**. $P$ values are calculated by unpaired two-sided $t$-test in **a**, **e**–**g**.

assay and confirmed QKI indeed bound to the putative QRE in *NDUFB2* pre-mRNA (Supplementary Fig. 2d). Moreover, *circNDUFB2* could be upregulated by QKI overexpression in NSCLC cells (Supplementary Fig. 2e). These data indicate that QKI binds to introns flanking the *circNDUFB2*-forming exons of *NDUFB2* pre-mRNA to promote *circNDUFB2* formation.

**circNDUFB2 inhibits NSCLC progression**. To study the role of *circNDUFB2* in NSCLC progression, we tested the endogenous expression of *circNDUFB2* in eight cell lines (Supplementary Fig. 3a). Furthermore, we constructed *circNDUFB2* overexpression plasmid (Supplementary Fig. 3b), and confirmed *circNDUFB2* was overexpressed accurately and efficiently in NSCLC cells (Supplementary Fig. 3c, d). *circNDUFB2* overexpression did not alter the mRNA and protein levels of *NDUFB2* (Supplementary Fig. 3e, f). These results indicate that *NDUFB2* is unaffected by *circNDUFB2*. Next, northern blotting analysis showed that exon circularization efficiency in the *circNDUFB2* overexpression vector is 56 and 53% in A549 cells and H1299 cells, respectively (Supplementary Fig. 3g). For *circNDUFB2* knockdown, four siRNAs specifically targeting the backsplice junction region were designed. si-*circNDUFB2*#1 and si-*circNDUFB2*#4 successfully silenced *circNDUFB2* and did not alter the mRNA levels of *NDUFB2* in NSCLC cells. si-*circNDUFB2*#2 and si-*circNDUFB2*#3 reduced the mRNA levels of *NDUFB2* (Supplementary Fig. 3h); it is likely due to the off-target effect caused by the partial similarity between *circNDUFB2* and *NDUFB2* mRNA sequences. Therefore, si-*circNDUFB2*#1 and si-*circNDUFB2*#4 were selected for the following loss-of-function assays of *circNDUFB2*.

Subsequently, in vitro studies showed that *circNDUFB2* overexpression significantly suppressed proliferation, migration, and invasion of NSCLC cells (Fig. 2a, c, e), and *circNDUFB2* knockdown remarkably promoted these phenotypes (Fig. 2b, d, f and Supplementary Fig. 3i, j). Moreover, in vivo experiments showed that *circNDUFB2* overexpression could markedly inhibit tumorigenicity and metastasis of NSCLC cells (Fig. 2g, h and Supplementary Fig. 3k). qRT-PCR analysis confirmed *circNDUFB2* overexpression in subcutaneous tumor tissues (Supplementary Fig. 3l). The tumor tissues with *circNDUFB2* overexpression had lower expression of Ki67 and PCNA than the vector group (Supplementary Fig. 3m). To further confirm the phenotypes of *circNDUFB2* overexpression that were caused by *circNDUFB2* instead of linear by-products from the pZW1-FCS-circNDUFB2, we constructed a linear-*NDUFB2* vector (pZW1-FCS-linearNDUFB2) by deleting the upstream complementary sequences in pZW1-FCS-circNDUFB2 plasmid. We found that pZW1-FCS-circNDUFB2 produced both *circNDUFB2* and linear-*NDUFB2*, whereas the pZW1-FCS-linearNDUFB2 only produced linear-*NDUFB2* (Supplementary Fig. 4a). Subsequently, in vitro studies showed that pZW1-FCS-linearNDUFB2 did not affect proliferation, migration, and invasion of NSCLC cells (Supplementary Fig. 4b, c). These data suggest that *circNDUFB2* may act as a suppressor in the progression of NSCLC.

**circNDUFB2 interacts with IGF2BP1/2/3 in NSCLC cells**. To test if *circNDUFB2* regulates targets as an miRNA sponge in NSCLC, we conducted RIP assay. Result showed that *ciRS-7* (a circRNA binding with AGO2)[28] was significantly enriched by the FLAG antibody, but not *circNDUFB2* (Supplementary Fig. 5a). This result suggests that *circNDUFB2* may not act as a miRNA sponge in NSCLC progression. To explore whether *circNDUFB2* exerted function via interacting with proteins, we conducted RNA pull-down assay to identify the proteins associated with it. qRT-PCR analysis confirmed that sense probe for *circNDUFB2* could

enrich *circNDUFB2* efficiently and specifically (Supplementary Fig. 5b). The precipitates in RNA pull-down assay were separated by SDS-PAGE. After silver staining, the sense-specific band at about 65 kDa (red arrow) was excised and analyzed using mass spectrometry (Supplementary Fig. 5c). We found the top three abundant proteins were IGF2BP2, IGF2BP1, and IGF2BP3, respectively (Supplementary Table 3). We then confirmed this result using western blot and RIP analysis (Fig. 3a, b). Notably, AGO2 was not detected in the precipitates (Supplementary Fig. 5d), which further confirmed that *circNDUFB2* did not act as a miRNA sponge. In addition, we performed RNA FISH-immunofluorescence analysis and found *circNDUFB2* co-localized with IGF2BPs in the cytoplasm (Fig. 3c).

It has been studied that KH domains of IGF2BPs are indispensable for the interactions between IGF2BPs and RNAs[20,29]. To explore whether KH domains of IGF2BPs are essential for interactions between IGF2BPs and *circNDUFB2*, we constructed IGF2BPs mutants with mutations of GxxG to GEEG in the KH domains as reported[20,29]. Mutations in the KH domains distinctly reduced interactions between IGF2BPs and *circNDUFB2* (Supplementary Fig. 5e, f). Results showed that *circNDUFB2* bound to KH domains of IGF2BPs. Because all three IGF2BPs preferentially bound to the "(U > C)GGAC" consensus[20], we identified two potential IGF2BP-binding regions ("CGGACU" and "UGGACA", yellow background) in *circNDUFB2* sequence (Supplementary Fig. 5g). We next conducted *circNDUFB2* mutant with mutations of "CGGACU" to "GCCUGA" and "UGGACA" to "ACCUGU" respectively (Supplementary Fig. 5g), and found that mutations in the *circNDUFB2* remarkably reduced interactions between IGF2BPs and *circNDUFB2* (Fig. 3d). Notably, these two IGF2BP-binding regions contain the "GGAC" $N^6$-methyladenosine (m$^6$A) core motif. IGF2BPs are m$^6$A-binding proteins and the capability of IGF2BPs binding to its targets is largely impaired by reducing the level of m$^6$A in mRNA[20]. The m$^6$A modification is installed by a multicomponent methyltransferase complex (MTC), which is composed of a core component *methyltransferase-like 3* (*METTL3*) and *methyltransferase-like 14* (*METTL14*) heterodimer. To explore whether interactions between circRNAs and IGF2BPs are regulated via an m$^6$A-dependent manner, we performed m$^6$A RNA immunoprecipitation (MeRIP) of *circNDUFB2*, and significant enrichment of m$^6$A in *circNDUFB2* was observed (Fig. 3e). Knockdown of *METTL3* and *METTL14* significantly reduced the level of m$^6$A modification in *circNDUFB2* and did not alter *circNDUFB2* level (Fig. 3e and Supplementary Fig. 5h, i). The interactions between *circNDUFB2* and IGF2BPs were remarkably impaired by *METTL3/14* knockdown (Fig. 3d). These data reveal that *circNDUFB2* physically interacts with three IGF2BPs in an m$^6$A-dependent manner.

**circNDUFB2 enhances IGF2BPs ubiquitination and degradation**. mRNA levels of IGF2BPs did not significantly change by *circNDUFB2* (Supplementary Fig. 5j, k), but protein levels of IGF2BPs were dramatically reduced in *circNDUFB2* overexpression and increased in *circNDUFB2* knockdown (Fig. 3f, g and Supplementary Fig. 5l). Moreover, *circNDUFB2* mutant did not affect protein levels of IGF2BPs (Fig. 3h). Therefore, we speculated that *circNDUFB2* may destabilize IGF2BP proteins through interaction. We found that *circNDUFB2* overexpression significantly reduced the levels of IGF2BP proteins, which could be restored by MG132, a specific proteasome inhibitor (Fig. 3i). *circNDUFB2* significantly increased the ubiquitination levels of IGF2BPs, but this effect was impaired by mutation of it (Fig. 3j). These results demonstrate that *circNDUFB2* reduces the stability of IGF2BPs via enhancing ubiquitin/proteasome-dependent degradation.

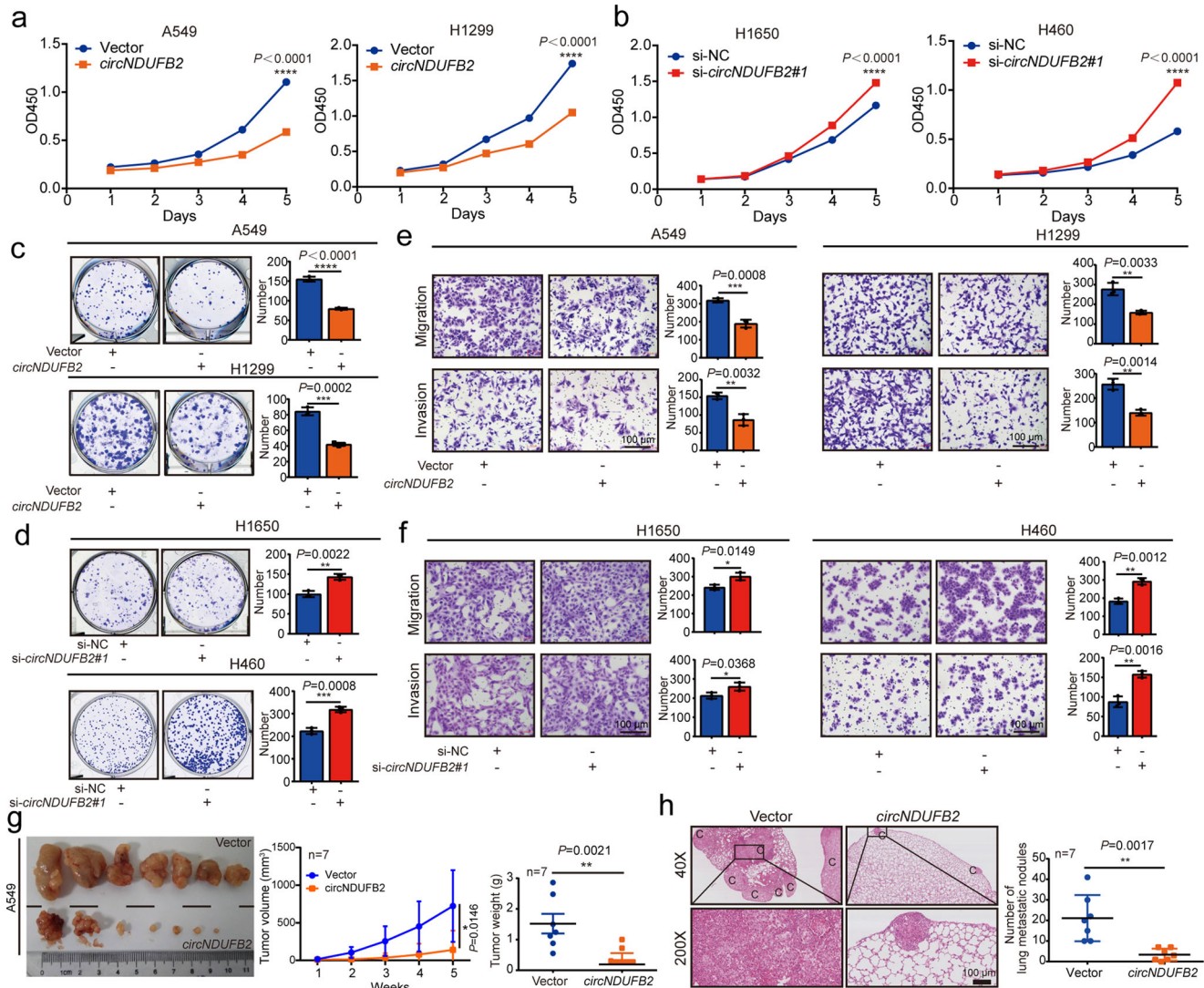

**Fig. 2 *circNDUFB2* inhibits proliferation and metastasis of NSCLC cells in vitro and in vivo. a**, **b** Cell proliferation assays for NSCLC cells with *circNDUFB2* overexpression or knockdown. $n = 5$ biologically independent samples. **c**, **d** Colony formation assays for NSCLC cells with *circNDUFB2* overexpression or knockdown. $n = 3$ biologically independent samples. **e**, **f** Migration and invasion assays for NSCLC cells with *circNDUFB2* overexpression or knockdown. $n = 3$ biologically independent samples. Scale bar $= 100 \, \mu m$. **g** The volume and weight of subcutaneous xenograft tumors ($n = 7$ mice per group). **h** Representative pictures of the lung metastases in nude mice by H&E staining ($n = 7$ mice per group). C cancer site. Scale bar $= 100 \, \mu m$. Data are presented as mean ± s.d. *P* values are calculated by unpaired two-sided *t*-test in **a**, **b** and **c**–**h**.

**TRIM25 is the E3 ligase that mediates IGF2BPs ubiquitination**. To further identify the E3 ligase for IGF2BPs ubiquitination, we performed mass spectrometry analysis for all precipitates in RNA pull down. Two E3 ligases, TRIM25 and HECTD3, were noticed among 156 potential interacting proteins (Supplementary Data 2). Using western blot assay, we observed that TRIM25 but not HECTD3 was specifically associated with *circNDUFB2* (Fig. 4a and Supplementary Fig. 6a). RIP assay further confirmed the association of *circNDUFB2* with TRIM25 (Supplementary Fig. 6b). RNA FISH-immunofluorescence analysis showed that *circNDUFB2* co-localized with the TRIM25 in the cytoplasm (Fig. 4b).

TRIM25 is a RNA-binding protein and belongs to the Tripartite Motif (TRIM) family of E3 ubiquitin ligases, which catalyzes the addition of polyubiquitin chains to its substrates for degradation[30–33]. We performed co-immunoprecipitation (Co-IP) analysis and results showed TRIM25 bound with all three IGF2BPs (Fig. 4c and Supplementary Fig. 6c). Immunofluorescence analysis indicated TRIM25 co-localized with all three

IGF2BP proteins in the cytoplasm (Fig. 4d). Interestingly, mRNA levels of IGF2BPs were not affected by TRIM25 (Supplementary Fig. 6d, e), but proteins levels of IGF2BPs were dramatically increased in TRIM25 knockdown and reduced in TRIM25 overexpression, respectively (Fig. 4e, f). Meanwhile, ubiquitination levels of IGF2BPs were dramatically increased in A549 cells with TRIM25 overexpression (Supplementary Fig. 6h–j). These results suggest that TRIM25 is the E3 ubiquitin ligase which interacts with IGF2BP proteins and degrades them via the ubiquitin–proteasome pathway in NSCLC cells.

The RNA-binding activity of TRIM25 is essential for its ubiquitin-ligase activity towards the substrate[34]. We then constructed a RNA-binding domain (RBD) deletion mutant of TRIM25 (lacking residues 470–508) with FLAG-tagged (termed TRIM25ΔRBD)[34], and confirmed that the deletion of RBD in TRIM25 significantly abolished the association between this protein and *circNDUFB2* (Supplementary Fig. 6f, g). These results indicate that TRIM25 binding to *circNDUFB2* dependents on its RNA-binding activity. Furthermore, TRIM25ΔRBD did not affect

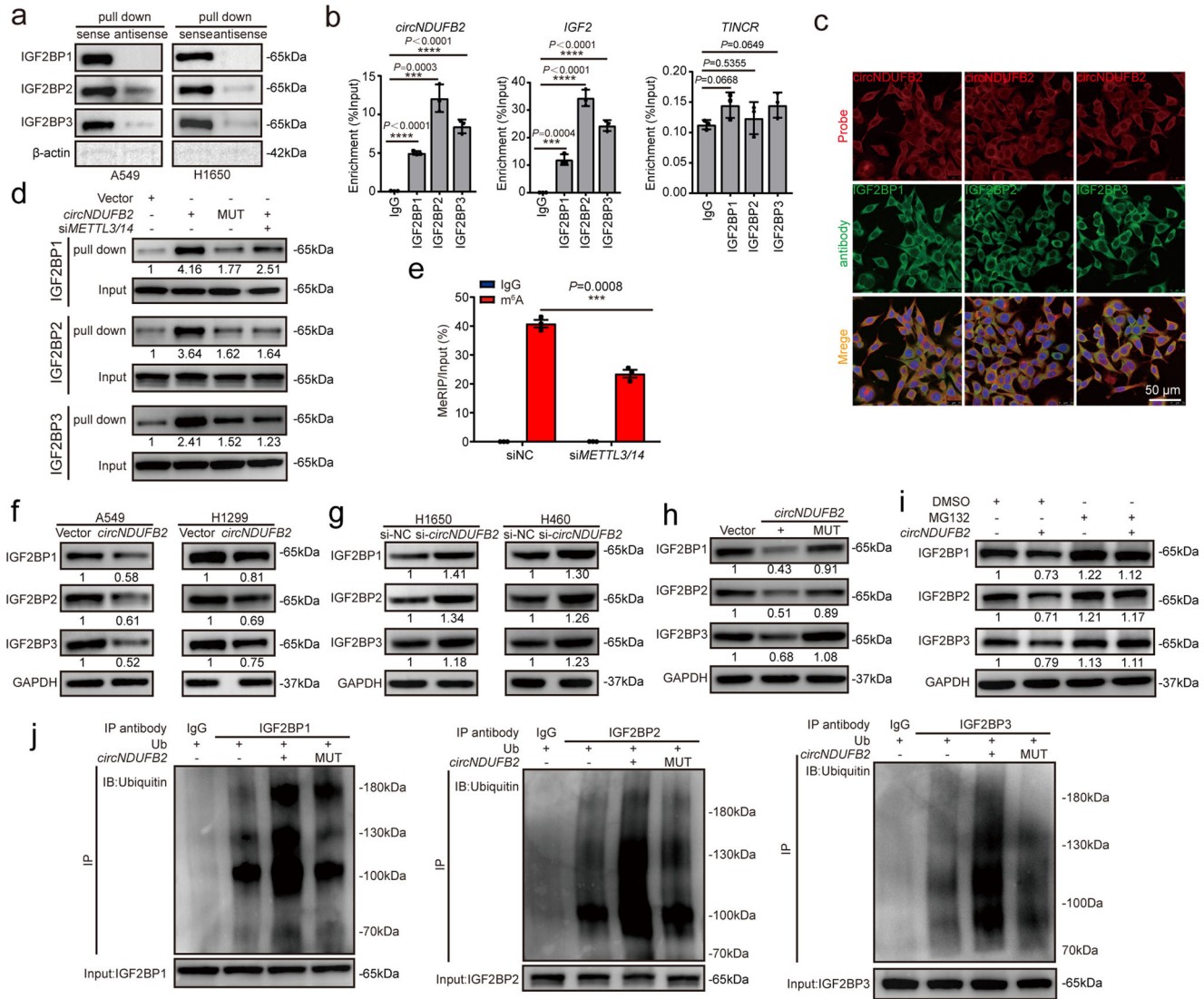

**Fig. 3 circNDUFB2 physically interacts with IGF2BPs and promotes ubiquitin/proteasome-mediated degradation of IGF2BPs. a** Interaction of IGF2BP1, IGF2BP2, and IGF2BP3 with *circNDUFB2*. β-ACTIN was used as a negative control. **b** Analysis for *circNDUFB2* enrichment. RIP assay was performed using the indicated antibodies in A549 cells. *IGF2* was used as a positive control and *TINCR* was used as a negative control. $n = 3$ biologically independent samples. **c** Co-localization of *circNDUFB2* (red) with IGF2BP proteins (green), respectively, in A549 cells. Scale bar = 50 μm. **d** RNA pull-down assays detected the interaction between *circNDUFB2* and IGF2BP proteins in the indicated group. **e** Gene-specific m6A qRT-PCR assay detected m6A modification of *circNDUFB2* in A549 cells with or without *METTL3/14* knockdown. $n = 3$ biologically independent samples. **f, g** Protein levels of IGF2BPs in NSCLC cells with *circNDUFB2* overexpression (**f**) or knockdown (**g**), respectively. **h** Protein levels of IGF2BPs in A549 cells with vector, *circNDUFB2*, or *circNDUFB2*-mutant transfection. **i** Protein levels of IGF2BPs in A549 cells with *circNDUFB2* overexpression treated with MG132 (20 μM) for 12 h. **j** Immunoprecipitation detected ubiquitination modification of IGF2BPs in A549 cells. Ub ubiquitin. Data are presented as mean ± s.d. *P* values are calculated by unpaired two-sided *t*-test in **b** and **e**. Two independent experiments were carried out with similar results in **b**, **c**, **d** and **f**–**j**.

protein levels of IGF2BPs (Fig. 4f), and no obvious effect on ubiquitination level of IGF2BPs was detected (Supplementary Fig. 6h–j), suggesting that the intact RBD of TRIM25 is necessary for efficient ubiquitination and degradation of IGF2BPs. Altogether, these results show that TRIM25 promotes IGF2BPs degradation via the ubiquitin–proteasome pathway in NSCLC cells, and RNA-binding activity of TRIM25 is essential for its roles in substrate ubiquitination.

**circNDUFB2 enhances interaction between IGF2BPs and TRIM25.** It has been shown that TRIM25 uses RNA as a scaffold for efficient ubiquitination of its targets[34]. We then explored whether ubiquitin-ligase activity of TRIM25 for IGF2BPs is

dependent on the presence of *circNDUFB2*. Loss of *circNDUFB2* displayed the notable weaken effects of TRIM25 on the ubiquitination and degradation of IGF2BPs (Fig. 4g and Supplementary Fig. 6k–m). To further verify our data and examine whether *circNDUFB2* acts as a scaffold to enhance the binding of TRIM25 with IGF2BPs, we performed Co-IP analysis. Remarkably, associations between TRIM25 and IGF2BPs were enhanced in A549 cells with *circNDUFB2* but not *circNDUFB2*-MUT overexpression (Fig. 4h–j and Supplementary Fig. 6n). Treatment with RNase A (a RNA endonuclease) but not RNase R (a RNA exonuclease) severely destroyed the interactions because of the resistance of *circNDUFB2* to RNA exonuclease (Fig. 4h–j and Supplementary Fig. 6n). Furthermore, the ubiquitination of IGF2BPs was significantly reduced upon *METTL3/14*

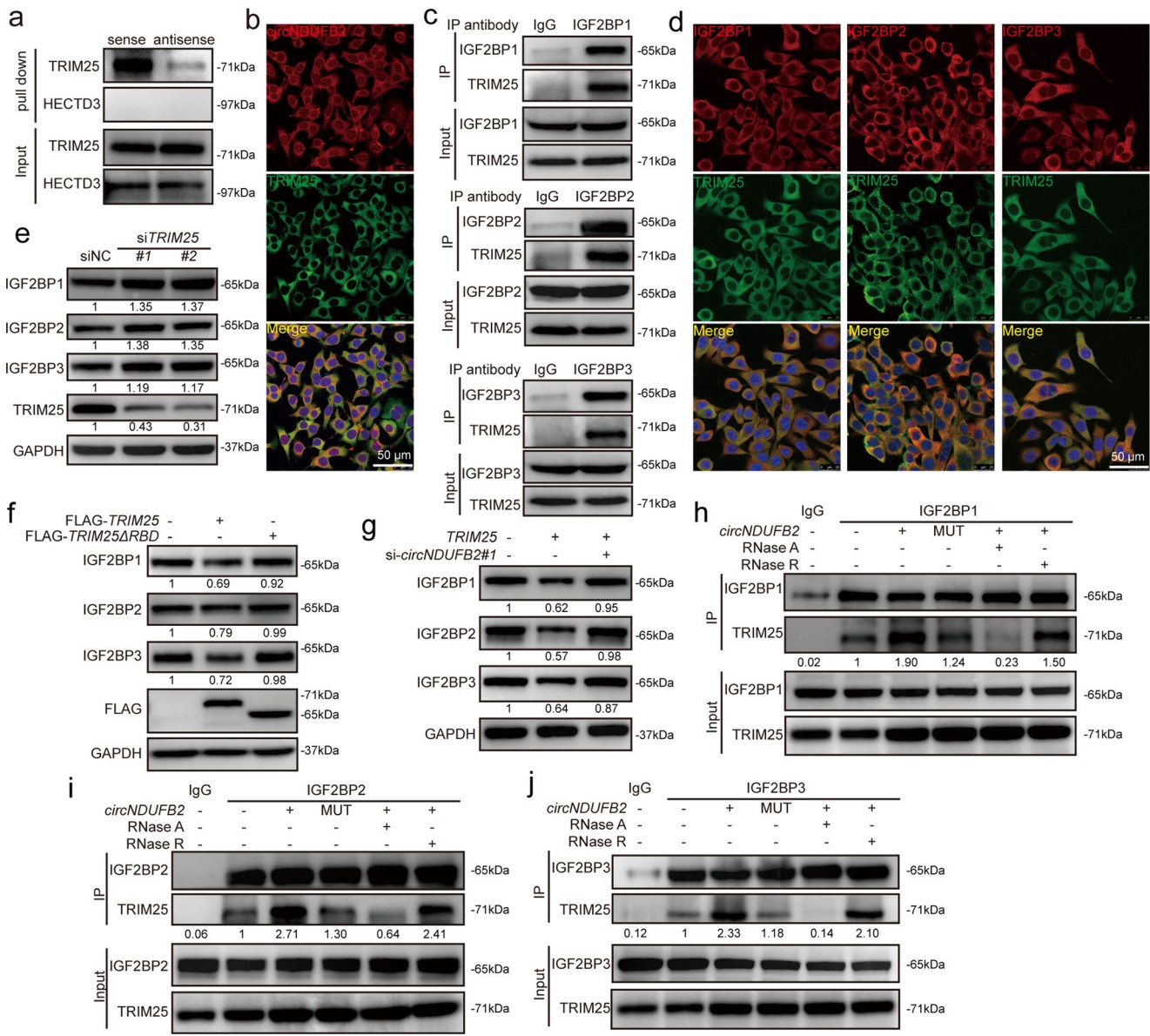

**Fig. 4 *circNDUFB2* acts as a scaffold to enhance the binding of IGF2BP proteins with TRIM25. a** Binding of *circNDUFB2* with TRIM25 or HECTD3 in A549 cells. **b** RNA FISH-immunofluorescence showed the co-localization of circNDUFB2 (red) with TRIM25 (green) in A549 cells. Scale bar = 50 μm. **c** Co-immunoprecipitation (Co-IP) showed the binding of TRIM25 with IGF2BPs in A549 cells. **d** Immunofluorescence detected the co-localization of IGF2BPs (red) with TRIM25 (green) in A549 cells. Scale bar = 50 μm. **e** Protein levels of IGF2BPs in A549 cells with TRIM25 knockdown. **f** Protein levels of IGF2BPs in A549 cells with TRIM25 or TRIM25-mutant (TRIM25ΔRBD) overexpression. **g** After TRIM25 overexpression, immunoblot detected protein levels of IGF2BPs in A549 cells with or without *circNDUFB2* knockdown. **h–j** Co-IP showed the binding of TRIM25 with IGF2BP proteins. Two independent experiments were carried out with similar results. RNase A (10 μg/ml), RNase R (100 U/ml).

knockdown (Supplementary Fig. 6o–q). Taken together, our results suggest that *circNDUFB2* functions as a scaffold to enhance the interaction between TRIM25 and IGF2BPs, subsequently facilitates the TRIM25-mediated ubiquitination and degradation of IGF2BPs.

**IGF2BPs are functional downstream mediator for *circNDUFB2*.** Because IGF2BPs are oncogenic proteins and *circNDUFB2* can reduce the stability of IGF2BPs, we hypothesized that IGF2BPs mediate the effects of *circNDUFB2* on NSCLC progression. We found that IGF2BPs overexpression significantly increased the abilities of migration and invasion as well as colony formation in A549 cells (Supplementary Fig. 7a, c). IGF2BPs

knockdown remarkably reduced the abilities of migration and invasion as well as colony formation in H1650 cells (Supplementary Fig. 7b, d). These results confirm IGF2BPs are tumor-promoting factors in NSCLC. Subsequently, we observed that IGF2BPs overexpression only partially restored the abilities of migration and invasion as well as colony formation reduced by *circNDUFB2* overexpression (Supplementary Fig. 7e, f), indicating that *circNDUFB2* exerts a tumor-suppressive role partially through degradation of IGF2BPs. Although *circNDUFB2*-MUT did not affect protein levels of IGF2BPs (Fig. 3h), it remained inhibitory effects for progression of NSCLC cells (Supplementary Fig. 7g, h). These data suggest that the inhibitory effects of *circNDUFB2*-MUT may result from other mechanisms of *circNDUFB2* in addition to degrading IGF2BPs.

***circNDUFB2* triggers immune defense response in NSCLC cells**. In order to investigate signaling pathways involved in *circNDUFB2*, unbiased transcriptome profiling was performed using RNA-sequencing (RNA-seq). Results showed that *circNDUFB2* overexpression affected expression levels of 934 genes, of which 743 genes were upregulated and 191 genes were downregulated (GEO accession number: GSE156607, https://www.ncbi.nlm.nih.gov/geo/query/acc.cgi?acc=GSE156607) (Fig. 5a and Supplementary Data 3). Gene ontology biological processes (GO_BP) and KEGG pathways enrichment analysis showed that *circNDUFB2* triggered immune defense and type I interferons (IFNs) signaling (Fig. 5b). Consistently, gene set enrichment analysis (GSEA) also showed that signaling pathways of target genes involved in immune responses (Fig. 5c). We confirmed a panel of immune gene expression was upregulated in NSCLC cells with *circNDUFB2* overexpression (Fig. 5d), whereas *circNDUFB2* knockdown reduced the levels of these gene in NSCLC cells (Fig. 5e). In addition, pZW1-FCS-linearNDUFB2 did not stimulate immune defense response in NSCLC cells (Supplementary Fig. 8a). These results indicated that overexpression of *circNDUFB2*, rather than its linear RNA fragment with the identical sequence, led to upregulation of immune genes. Enzyme-linked immunosorbent assays (ELISAs) confirmed that *circNDUFB2* overexpression dramatically increased the levels of CXCL10, CXCL11, CCL5, and IFNβ in cell culture supernatants, and *circNDUFB2* knockdown reduced the levels of these cytokines in cell culture supernatants (Fig. 5f, g). These results demonstrate that *circNDUFB2* elicits immune responses in NSCLC cells.

**RIG-I is essential for *circNDUFB2*-induced immune responses**. RIG-I-like receptor (RLR) family comprises retinoic acid-inducible gene-I (RIG-I) and melanoma differentiation-associated protein 5 (MDA5), as well as laboratory of genetics and physiology 2 (LGP2), recognizes viral RNAs in the cytoplasm and triggers innate immune responses against viral infection through the production of IFNs and proinflammatory cytokines[35–42]. Recently, it has been reported that exogenous circRNAs potently stimulate immune signaling which is mediated by RIG-I (ref. [22]). Our data from qRT-PCR analysis showed that RIG-I but not MDA5 knockdown abrogated immune responses induced by *circNDUFB2* overexpression (Fig. 6a and Supplementary Fig. 8b). We also noticed RIG-I was upregulated by *circNDUFB2* (Fig. 6a and Supplementary Fig. 8b). To investigate association between *circNDUFB2* and RIG-I, we next performed RIP assay and RNA FISH-immunofluorescence experiments. Results showed that *circNDUFB2* bound with RIG-I (Fig. 6b), and they were co-localized in the cytoplasm (Supplementary Fig. 8c). After *circNDUFB2* overexpression, the expression of immune genes in A549 cells with or without RIG-I knockdown was further analyzed by western blot. Results showed that protein levels of RIG-I, IRF7, IFNβ, and TNF were significantly upregulated by *circNDUFB2*, and protein levels of P65 and IRF3 did not affect by *circNDUFB2* (Fig. 6c). Meanwhile, p-P65, p-IRF3, p-STAT1, and p-STAT2 were significantly increased by *circNDUFB2* (Fig. 6c). Data also showed that *circNDUFB2* promoted translocation of IRF3 and P65 into the nucleus (Fig. 6d). Furthermore, immunofluorescence analysis confirmed that *circNDUFB2* promoted phosphorylation of IRF3, and subsequently triggered it to translocate into the nucleus (Supplementary Fig. 8d). More importantly, RIG-I knockdown significantly abolished the upregulation of protein levels or phosphorylation levels of these genes as well as nuclear translocation of IRF3 and P65 induced by *circNDUFB2* in A549 cells (Fig. 6c, d and Supplementary Fig. 8d). Next, we performed ELISAs to measure the level of cytokines in cell culture

supernatants and found that RIG-I knockdown significantly abrogated the induction of CXCL10, CXCL11, CCL5, and IFNβ (Fig. 6e). To explore whether RIG-I binds *circNDUFB2* at m6A modification sites, we performed RNA pull-down assay in NSCLC cells with *circNDUFB2* or *circNDUFB2* mutant (Fig. S5g). We noticed that mutation of m6A modification site in *circNDUFB2* did not affect its interaction with RIG-I (Supplementary Fig. 8e). Consistently, the expression level of anti-tumor immune-related genes has no significant difference between *circNDUFB2* and *circNDUFB2* mutant (Supplementary Fig. 8f). The above results suggest a pivotal role for RIG-I in mediating immune responses of *circNDUFB2* in NSCLC cells, and *circNDUFB2*-elicited immune response is not dependent on m6A modification in *circNDUFB2*.

In the absence of RNA ligand, RIG-I adopts an auto-repressed conformation that prevents the N-terminal caspase recruitment domains (CARDs) from signaling[36,41,43]. Therefore, we hypothesized that *circNDUFB2* may activate RIG-I through facilitating its CARDs release and keeping RIG-I in an active form. We performed Co-IP to detect interaction between CARDs (FLAG-CARD) and helicase domain (HA-ΔCARD) of RIG-I. Results showed that *circNDUFB2* inhibited interaction between CARDs and helicase domain of RIG-I (Fig. 6f). Furthermore, *circNDUFB2* enhanced the interaction of CARDs with mitochondrial anti-viral signaling protein (MAVS) (Fig. 6f). These results suggest that *circNDUFB2* may activate RIG-I through destabilizing intramolecular interaction between CARDs and its helicase domain, and keeping RIG-I in an active form and subsequently eliciting the activation of *RIG-I-MAVS* signaling cascades.

**Immune responses of *circNDUFB2* inhibit NSCLC progression**. RIG-I knockdown partially restored the abilities of colony formation as well as migration and invasion reduced by *circNDUFB2* overexpression (Supplementary Fig. 9a, b). However, the inhibitory effects induced by *circNDUFB2* mutant could be completely restored by RIG-I knockdown (Supplementary Fig. 9c, d). These data suggest that besides IGF2BPs, RIG-I-mediated immune response also plays important roles in inhibitory effects of *circNDUFB2* on NSCLC progression.

To explore whether *circNDUFB2* has potential to stimulate immune responses and then recruit immune cells into tumor microenvironment (TME), LLC1 (LL/2, a murine lung carcinoma cell line) cells with or without *circNDUFB2* overexpression were delivered into C57BL/6 mice by subcutaneous injection. Three weeks later, we found *circNDUFB2* overexpression markedly inhibited the tumorigenicity of LLC1 cells in vivo (Fig. 6g), and the level of IFNβ in serum was significantly increased in the mice with *circNDUFB2*-overexpressed LLC1 cells (Fig. 6h). Moreover, the infiltration of CD8+ T cells and DCs was detected in the tumor tissues dissected from these mice, and *circNDUFB2* overexpression significantly increased the frequency of CD8+ T cells and DCs in the TME (Fig. 6i and Supplementary Fig. 9e, f). Finally, we analyzed correlations between *circNDUFB2* and *RIG-I* or *IFNβ* in the tumorous tissues of 52 NSCLC patients. Results showed that *circNDUFB2* was positively correlated with *RIG-I* and *IFNβ* (Fig. 6j). The above results indicate that immune response of *circNDUFB2* mediated by RIG-I inhibits tumor progression.

***circNDUFB2* inhibits NSCLC progression in a dual-role pattern**. To further verify whether *circNDUFB2* inhibits NSCLC progression via destabilizing IGF2BPs and activating innate immunity, we performed colony formation assays as well as migration and invasion assays to measure the restoring effects of IGF2BPs and RIG-I on A549 cells. Results displayed that

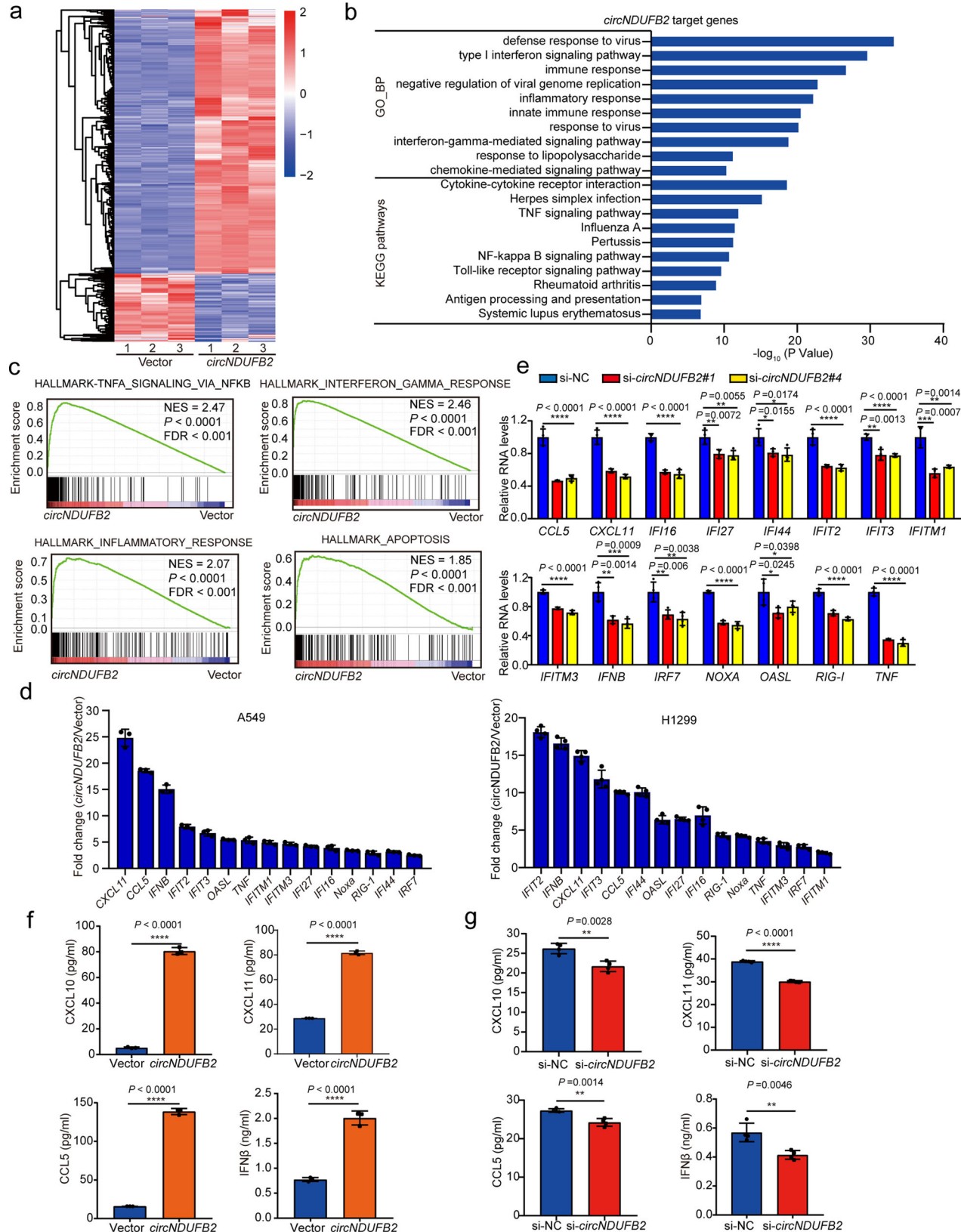

IGF2BPs overexpression together with RIG-I knockdown could significantly and completely restore inhibitory effects of *circNDUFB2* overexpression on A549 cells (Supplementary Fig. 9g, h). These data suggest that both IGF2BPs and RIG-I are crucial mediators for the inhibitory effects of *circNDUFB2* on NSCLC progression. Finally, we calculated the copy numbers of IGF2BPs,

RIG-I and *circNDUFB2* in NSCLC cells by absolute quantification[8,43] (Supplementary Fig. 9i, j). We measured 2221 or 1842 copies of IGF2BPs per A549 cell or per H1299 cell, respectively. *circNDUFB2* calculation suggest that each A549 cell contained 151 copies of *circNDUFB2* and each H1299 cell contained 206 copies of *circNDUFB2*. The long half-life of

**Fig. 5 *circNDUFB2* provokes cellular immune responses. a** Nine hundred and thirty-four dysregulated target genes in A549 cells with *circNDUFB2* overexpression (*circNDUFB2*) versus control (Vector) were identified by RNA-seq analysis. Fold change > 2.0 or fold change < −2.0, *P* value < 0.05 as significant change. **b** Enrichment analysis for representative GO_BP and KEGG pathways in *circNDUFB2* target genes. **c** GSEA analysis for the *circNDUFB2* overexpression group (*circNDUFB2*) compared to the control group (Vector). NES normalized enrichment score, FDR false discovery rate. *P* values are calculated by permutation test. **d** qRT-PCR measured the fold change of indicated mRNAs in A549 and H1299 cells with or without *circNDUFB2* overexpression. *n* = 3, 4 biologically independent samples, respectively. **e** Expression levels of indicated mRNA in H1650 cells with *circNDUFB2* knockdown. *n* = 4 biologically independent samples. **f, g** ELISA measured the levels of CXCL10, CXCL11, CCL5, and IFNβ in the supernatants of A549 cells with *circNDUFB2* overexpression (**f**) or H1650 cells with *circNDUFB2* knockdown (**g**). *n* = 3, 4 biologically independent samples, respectively. Data are presented as mean ± s.d. in **d**–**g**. *P* values are calculated by unpaired two-sided *t*-test in **a**, and **e**–**g**.

*circNDUFB2* makes it work longer as a functional molecule and each *circNDUFB2* contains two IGF2BP-binding motifs, although the copy number of *circNDUFB2* is far less than IGF2BPs, *circNDUFB2* still has the capacity to bind a portion of IGF2BPs and thus promote the degradation of IGF2BPs. Moreover, results also showed each A549 cell and each H1299 cell contained 50 and 286 copies of RIG-I, respectively. Considering that RIG-I mediated anti-tumor immunity and *circNDUFB2* could upregulate RIG-I expression (Fig. 6a, c), the upregulation of RIG-I by *circNDUFB2* might constitute a positive feedback loop to elicit robust anti-tumor immunity. These results suggest that the downregulation of *circNDUFB2* in NSCLC not only enhance IGF2BPs stability but also contribute to immune evasion.

## Discussion

circRNAs are first discovered more than 40 years ago; however, they are usually considered as by-products generated by aberrant splicing events with little functional potential[7]. With the advances of high-throughput RNA-sequencing and circRNAs-specific bioinformatics algorithms, a large number of circRNAs have been identified in eukaryotes, and some of them have been implicated in tumor progression[11,12]. In this study, we identify *circNDUFB2* is frequently downregulated in NSCLC tissues, and decrease of QKI contributes to the downregulation of *circNDUFB2*. *circNDUFB2* not only promotes IGF2BPs ubiquitination degradation by forming TRIM25/circNDUFB2/IGF2BPs ternary complex, but also triggers cellular immune responses by activating RIG-I. We first report that *circNDUFB2* inhibits NSCLC progression via destabilizing IGF2BPs and activating immune responses.

Increasing evidences indicate that expression level of circRNA was associated with clinicopathologic features in tumor patients[13,18]. We found decreased *circNDUFB2* in NSCLC tissues was significantly correlated with larger tumor size and lymph node metastasis as well as higher stage in NSCLC patients (Supplementary Table 1). However, we noticed most of the patients (36/52) were nonsmokers in our study. The smoking status in our cohort was different from what is typically seen in lung cancer that 90% of male and 79% of female lung cancers were attributable to tobacco smoking[2,44]. Therefore, whether expression of *circNDUFB2* is associated with smoking status in NSCLC patients needs further investigation by expanding the sample size of the cohort. Moreover, oncogenic driver mutations or gene rearrangements are commonly observed in NSCLC. Variant allele frequencies for somatic mutations have shown that mutation in the epidermal growth factor receptor (*EGFR*) gene is usually present within NSCLC patients. Frequency of EGFR-activating mutation is 27% in LUAD and 9% in LUSC.[2] However, the correlation between EGFR mutational status and *circNDUFB2* is unknown and needs to be studied in the future.

In NSCLC, most studies focus on the role of circRNAs as miRNA sponges[17,18]. In this study, we demonstrated that *circNDUFB2* exerted its function not through ceRNA mechanism,

but through reducing stability of IGF2BPs by forming TRIM25/ circNDUFB2/IGF2BPs ternary complex (Figs. 3 and 4 and Supplementary Figs. 5–7). IGF2BPs promote growth and metastasis of various types of cancer via enhancing mRNA stability or translation of oncogenic factors[19,20,45–47]; however, little is known about the degradation of IGF2BP proteins. Here, our findings first expose that *circNDUFB2* functions as a scaffold to facilitate the TRIM25-mediated ubiquitination of IGF2BPs. We found TRIM25 exerted its ubiquitin-ligase activity dependent on its RNA-binding activity (Fig. 4f and Supplementary Fig. 6h–j), and *circNDUFB2* was necessary for efficient ubiquitination of IGF2BPs by TRIM25 (Fig. 4g and Supplementary Fig. 6k–m). Our data provide a previously unrecognized mechanism for IGF2BPs degradation and novel evidences for circRNAs participating in protein metabolism.

$N^6$-methyladenosine is the most abundant and reversible internal modification in mRNAs and circRNAs[48,49]; recent studies have shown that m6A modification participates in circRNAs metabolism[50,51]. As m6A reader, IGF2BPs bind to thousands of target RNAs, and the capability of IGF2BPs binding to its targets can be affected by the level of m6A modification in RNAs[20]. In this study, we revealed that m6A modification enriched in *circNDUFB2* (Fig. 3e). The level of m6A modification in *circNDUFB2* impacted the strength of *circNDUFB2* binding to IGF2BPs (Fig. 3d, e), and then affected ubiquitin-ligase activity of TRIM25 for IGF2BPs (Supplementary Fig. 6o–q). Our results develop the knowledge about the effects of m6A modification on circRNAs function.

Recent study suggests that exogenous circRNA acts as a vaccine adjuvant to increase the efficacy of the vaccine and induces antigen-specific T cell activation[23]. These data denote the function of circRNA in activating cellular immune response, whereas how circRNA triggers immune responses in cancer cells need to further explore. Here, we exhibited *circNDUFB2* triggered immune defense response in NSCLC cells (Fig. 5). Our results showed *circNDUFB2* was recognized by RIG-I but not MDA5 (Fig. 6a and Supplementary Fig. 8b). Probably, it is due to the fact that the length of *circNDUFB2* is 249 nt, and RIG-I is preferentially activated by short dsRNAs (<300 bp), whereas MDA5 is preferentially activated by longer dsRNAs (>4 kbp)[52]. *circNDUFB2* bound with RIG-I and activated it through destabilizing intramolecular interaction between CARDs and its helicase domain, and subsequently enhanced interaction of CARDs with MAVS (Fig. 6f). After activation of *RIG-I–MAVS* pathway, a phosphorylation cascade allowed signal transmission to lead to activation of IRF3 and NF-κB, and they translocated into the nucleus to drive transcription of IFNs and chemokines (Fig. 6c–e and Supplementary Fig. 8d). Immune cells, such as CD8+ T cells and DCs, were recruited into the TME by the secretion of chemokines (Fig. 6i and Supplementary Fig. 9e, f). We found that IFNs activated the *Jak/STAT* pathway in an autocrine and paracrine manner and drove transcription of interferon-stimulated genes (ISGs) (Fig. 6a, c). Both RIG-I and IRF7 belong to ISGs that can be induced by IFNs[36,53]. RIG-I and IRF7 were indispensable

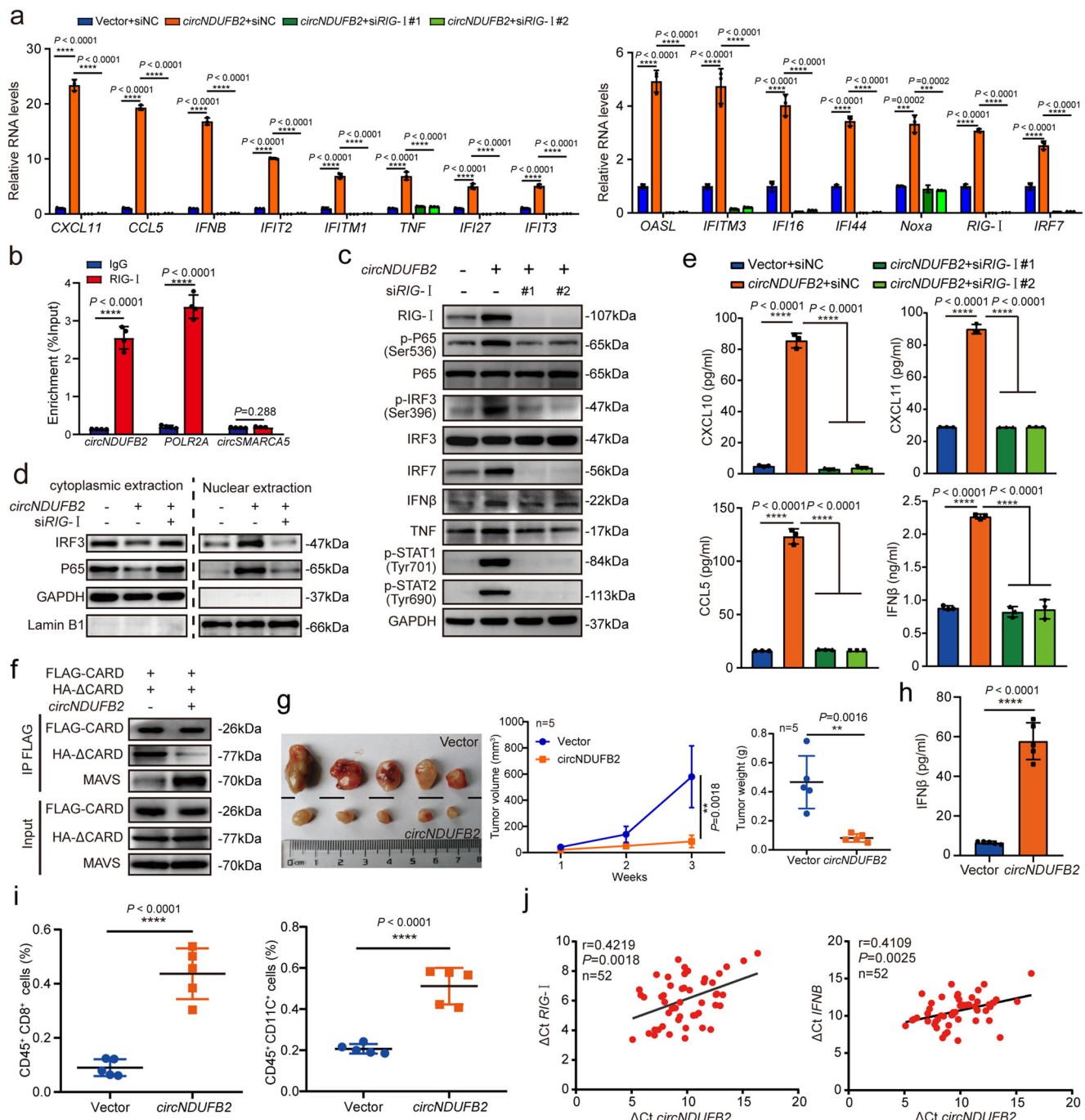

**Fig 6 *circNDUFB2* activates downstream signaling of RIG-I. a** Fold change of indicated mRNAs in A549 cells with or without RIG-I knockdown. $n = 3$ biologically independent samples. **b** Analysis for *circNDUFB2* enrichment. RIP assay was performed using RIG-I antibody in A549 cells. *POLR2A* was used as a positive control and *circSMARCA5* was used as a negative control. $n = 4$ biologically independent samples. **c** Protein levels in A549 cells with *circNDUFB2* overexpression or RIG-I knockdown. **d** Protein levels of IRF3 and P65 in the nuclear and cytoplasmic fractions in the indicated cells. **e** Levels of CXCL10, CXCL11, CCL5, and IFNβ in the supernatants of A549 cells with *circNDUFB2* overexpression or RIG-I knockdown. $n = 3$ biologically independent samples. **f** Interactions between CARDs of RIG-I with helicase domain of RIG-I and MAVS in A549 cells. FLAG-CARD, CARDs of RIG-I with FLAG-tagged; HA-ΔCARD, helicase and CTD of RIG-I with HA-tagged. **g** The volume and weight of subcutaneous xenograft tumors ($n = 5$ mice per group). **h** Levels of IFNβ in the serum of mice ($n = 5$ mice per group). **i** The percentage of CD8[+] T cells and DCs in subcutaneous xenograft tumors. $n = 5$ tumors per group. **j** Correlation analysis for *circNDUFB2* and *RIG-I* or *IFNβ* in the tumorous tissues of 52 NSCLC patients. ΔCt values were normalized according to *GAPDH*. $n = 52$ biologically independent NSCLC tissues. R represents the Pearson correlation coefficient. Data are presented as mean ± s.d. P values are calculated by unpaired two-sided t-test in **a**, **b**, **e**, **g–i**. P values in **j** are calculated by F-test. Two independent experiments were carried out with similar results in **c**, **d**, **f**.

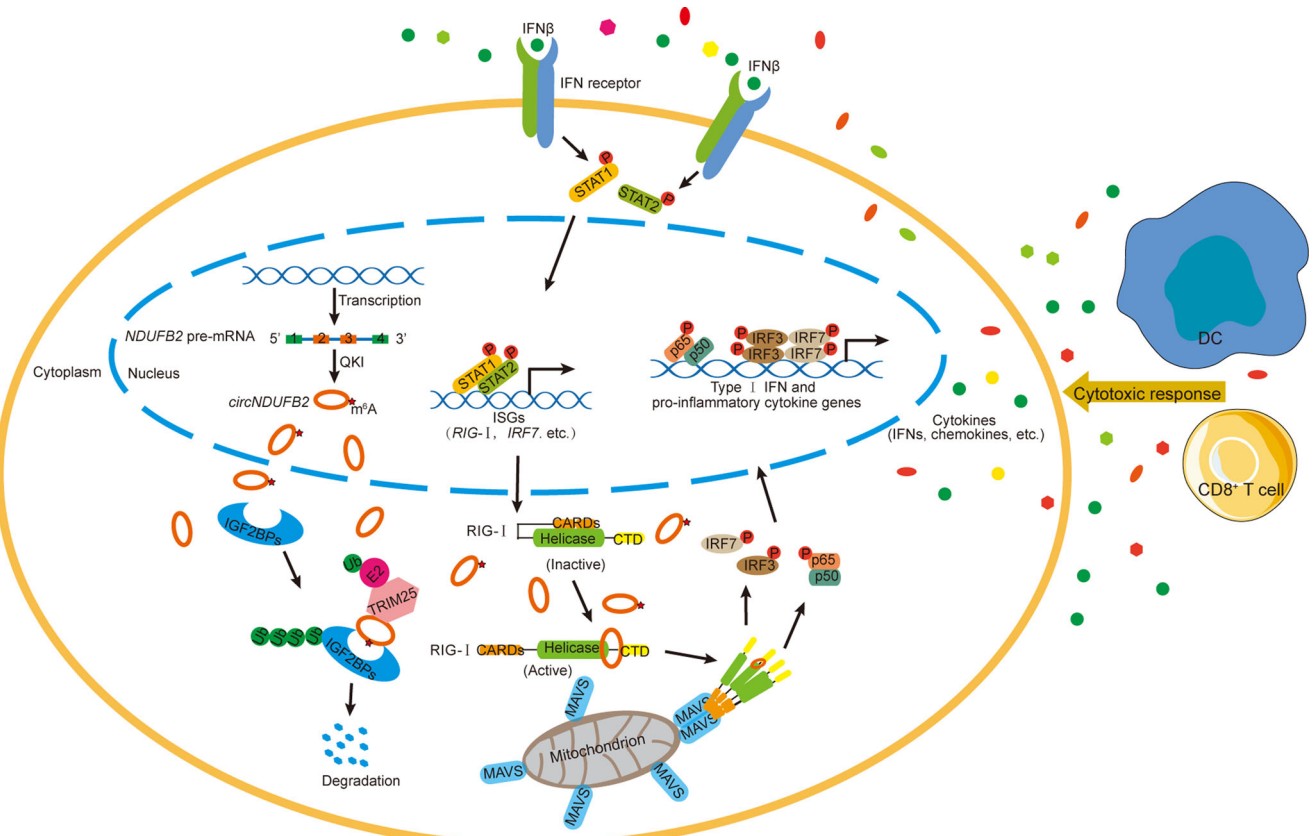

**Fig. 7 Proposed working model for inhibitory effects of *circNDUFB2* on NSCLC progression.** *circNDUFB2* derives from the Exons 2–3 of *NDUFB2* gene by backsplicing in the nucleus and exports to the cytoplasm. *circNDUFB2* not only facilitates ubiquitination degradation of IGF2BPs by forming a TRIM25/ circNDUFB2/IGF2BPs ternary complex but also recruits immune cells into TME by activating the *RIG-I–MAVS* pathway, thus leading to inhibition of tumor growth.

in immune signaling transmission for *circNDUFB2*, and the induction of RIG-I and IRF7 by *circNDUFB2* may constitute a positive feedback loop and provoke robust cellular immune responses (Fig. 7). Based on these in vitro and in vivo experiments, we first report that *circNDUFB2* participates in the infiltration of immune cells into TME and regulation of NSCLC progression. Our findings broaden the understanding of mechanism in circRNAs activating anti-tumor immunity.

In conclusion, *circNDUFB2* participates in degradation of IGF2BPs and it is a previously unknown natural agonist for RIG-I in cells. We broaden the knowledge of circRNAs action in NSCLC progression. This study implies that *circNDUFB2*, as a potent agonist of RIG-I, may have attractive translational potential for the immunotherapy of NSCLC.

## Methods

**Tissues**. In total, 55 paired primary NSCLC tumorous tissues (T) and adjacent nontumorous tissues (N) were collected from patients who had undergone surgery at Shanghai Chest Hospital (Shanghai, China). All paired samples of tumorous and nontumorous tissues were confirmed by two pathologists independently. Among them, three paired samples of tumorous and adjacent nontumorous tissues were used for microarray analysis, and other 52 paired samples for qRT-PCR verification. All samples were all stored at −80 °C until use. The detailed clinicopathological characteristics are described in Supplementary Tables 1 and 2. All tissue specimens were collected from July 2013 to September 2014 with the consent of patients and approved by Ethics Committee of the Shanghai Chest Hospital.

**Mice and cell lines**. Female BALB/c nude mice (BALB/cAnNShjhnu) and C57BL/ 6 mice (C57BL/6JShjh) (6–8 weeks old) were purchased from Shanghai Jihui Laboratory Animal Care Co., Ltd (Shanghai, China) and maintained under SPF conditions in a controlled environment of 20–22 °C, with a 12/12 h light/dark cycle, 50–70% humidity, and food and water provided ad libitum. The small animal

euthanasia equipment was used for animal euthanasia. Mice were put into the euthanasia chamber filling with 99.9% of $CO_2$ gas for 10 min. All animal experiments were performed under approval by the Shanghai Medical Experimental Animal Care Commission.

All of human NSCLC cell lines (A549, H1299, HCC827, H1975, H1703, H460, and H1650), human bronchial epithelial cell line (BEAS-2B), and murine lung carcinoma cell line (LLC1) were purchased from the American Type Culture Collection (ATCC). A549 cells were cultured in F-12K medium (Gibco) with 10% FBS (Gbico). H1299, HCC827, H1975, H1703, H460, and H1650 cells were cultured in RPMI-1640 medium (Gibco) with 10% FBS. BEAS-2B cells were cultured in BEGM medium (Gibco) with 10% FBS. LLC1 cells were cultured in DMEM medium (Gibco) with 10% FBS. They were all cultured at 37 °C with 5% $CO_2$.

**Microarray analysis**. The sample preparation and microarray hybridization were performed based on the Arraystar's standard protocols. Briefly, total RNAs isolated from each sample were amplified and transcribed into fluorescent cRNA utilizing random primer according to Arraystar's Super RNA Labeling protocol (Arraystar Inc.). The labeled cRNAs were hybridized onto the Arraystar Human Circular RNA Microarray V1.0 (6x7K, Arraystar), which contains 5396 probes specific for human circular RNAs backsplice junction region. After having washed the slides, the arrays were scanned by the Axon GenePix 4000B microarray scanner. Scanned images were then imported into GenePix Pro 6.0 software (Axon) for grid alignment and data extraction. Quantile normalization and subsequent data processing were performed using the R software package. Significant differential expressed circRNAs were screened by fold change > 2 or < −2 and P value < 0.05.

**RT-PCR and qRT-PCR**. RNA was reverse-transcribed using HiScript II Q RT SuperMixfor qPCR (+gDNA wiper) (Vazyme, Nanjing, China). AceQ qPCR SYBR Green Master Mix (Vazyme, Nanjing, China) was used for qRT-PCR. The circRNA and mRNA levels were normalized by *GAPDH*. Oligonucleotides sequences are listed in Supplementary Data 4.

**RNase R treatment**. Two micrograms of total RNA was incubated 30 min at 37 °C in the absence or presence of 5 U/μg RNase R (Epicentre Technologies, Madison,

WI, USA), and the resulting RNA was subsequently purified by RNeasy MinElute Cleaning Kit (Qiagen), and then analyzed by qRT-PCR.

**Actinomycin D assays**. A549 cells and H1650 cells were seeded in six-well plates ($5 \times 10^5$ cells per well). Twenty-four hours later, cells were exposed to 2 μg/ml Actinomycin D (Sigma) and collected at indicated time points. The RNA stability was analyzed using qRT-PCR and normalized to the values measured in the mock treatment group (the 0 h group).

**Vector construction and cell transfection**. A sketch structure of *circNDUFB2* overexpression and empty vector plasmid construction were shown (Supplementary Fig. 3b). Briefly, for *circNDUFB2* overexpression version, *circNDUFB2* with its flanking introns (20 nt upstream and downstream intron sequence, respectively) was amplified from cDNAs reverse-transcribed from 293 T cells, and PCR product was cloned into the *XhoI*/*AflII*-digested pZW1-FCS-circRNA vector (Addgene: 73449, a gift from Ling-Ling Chen, Shanghai Institute of Biochemistry and Cell Biology) using the Peasy-Uni Seamless Cloning and Assembly Kit (Transgen, Beijing, China). For the mutant version of *circNDUFB2*, the sequence of "CGGACU" or "UGGACA" (in yellow background) was replaced with "GCCUGA" or "ACCUGU" (in blue background), respectively, using the Fast Mutagenesis Kit V2 (Vazyme, Nanjing, China) (Supplementary Fig. 5g). For the empty vector version, 20 nt upstream and downstream flanking intron sequence of *circNDUFB2* was synthesized and cloned into the *XhoI*/*AflII*-digested pZW1-FCS-circRNA vector as mentioned above.

The transfection of siRNAs and plasmids was performed using the Lipofectamine 3000 kit (Invitrogen) according to the manufacturer's instructions. The siRNA sequences are listed in Supplementary Data 4.

**In vitro cell behavior assays**. For the cell proliferation assay, $1 \times 10^3$ cells were seeded in 100 μl complete culture media in 96-well plates for various time periods. Cell Counting Kit-8 assay (Dojindo Laboratories, Kumamoto, Japan) was performed to measure cell viability according to the manufacturer's instructions. For colony formation assay, 500 cells were seeded in 2 ml complete culture media in six-well plates for 10 days. After 10 days, colonies were stained using Crystal Violet and counted. For migration and invasion assays, transwell filter champers (Costar, Corning, NY) with or without Matrigel (BD Biosciences) were used according to the manufacturer's instructions. Cells migrated through the membrane were fixed, stained, and counted under a light microscope.

**In vivo tumorigenesis and metastasis assays**. For xenograft experiments, 6–8-week-old female BALB/c nude mice were used. A549 cells were transfected with the pZW1-FCS-circNDUFB2 plasmid or pZW1-FCS-Vector plasmid, and selected with G418 (800 μg/ml) for 4 weeks, and then $2 \times 10^6$ A549 cells were subcutaneously injected into the right flank of each mouse. When a tumor was palpable, it was measured using a caliper every week, and its volume was calculated according to the formula volume = length × width$^2$ × 0.5. The early end point was defined as the time at which a progressively growing tumor volume reached 1500 mm$^3$. For the tail vein metastases model, $1 \times 10^6$ A549 cells were injected into the tail vein of nude mice. Mice were euthanized when they experienced lost more than 15% of their total body weight or failed to thrive. The lung tissues were dissected and fixed with 4% phosphate-buffered neutral formalin. Lung tissues were analyzed by H&E staining.

For C57BL/6 mice xenograft experiments, 6–8-week-old female C57BL/6 mice were used. LLC1 cells were transfected with the pZW1-FCS-circNDUFB2 plasmid or pZW1-FCS-Vector plasmid, and selected with G418 (800 μg/ml) for 4 weeks, and then $2 \times 10^5$ LLC1 cells with or without *circNDUFB2* overexpression were injected into the right flank of each mouse. The mice were euthanized when the tumor volume reached 1500 mm$^3$. The tumors were dissected, and lymphocyte infiltration was detected by flow cytometry and IHC.

**RNA FISH-immunofluorescence microscopy**. To detect co-localization of *circNDUFB2* with indicated proteins, A549 cells were fixed, permeabilized, and pre-hybridized. Subsequently, hybridization using Cy-3-conjugated *circNDUFB2* probes was performed at 37 °C in the dark overnight, and cells were rinsed in SSC buffer at 42 °C. Then the cells were incubated with the blocking buffer (PBST containing 5% bovine serum albumin) for 30 min at room temperature. Subsequently, the cells were incubated with primary antibody (1:200 dilution) for 1 h at room temperature, which was followed by a reaction with Alexa Fluor 594- or 488-conjugated secondary antibodies (1:200 dilution; Cell Signaling Technology) and DAPI (Vector Laboratories) for 30 min. The images were obtained using a confocal microscope. Antibody information is listed in Supplementary Table 4.

**Western blot analysis**. An equal amount of total protein lysates (20 μg) was separated by 6–15% SDS-PAGE and transferred onto a PVDF membrane. After blocking for nonspecific binding, the membranes were incubated with primary antibody (1:1000 dilution) overnight at 4 °C, followed by an incubation with secondary antibody (1:5000 dilution; Bioword) 1 h at room temperature. Bands were

detected by a Bio-Rad ChemiDoc XRS system. Antibody information is listed in Supplementary Table 4.

**Measurement of protein molecular number**. A serial dilution of purified recombinant protein was used for western blot to generate the standard curve. The mass of interest protein in a known number of cells was quantitated from the standard curve. Western blot signal intensities were quantified using the ImageJ program. The protein molecular weight was calculated by Protein Molecular Weight Calculator from the following website (http://endmemo.com/bio/promw.php). Purified recombinant RIG-I and IGF2BP2 proteins were from Abcam, and IGF2BP1 and IGF2BP3 proteins were from RayBiotech.

**Measurement of circRNA copy number**. A serial dilution of purified RT-PCR products of *circNDUFB2* was used for qRT-PCR to generate the standard curve. The copy number of the diluted DNA template was calculated by the DNA/RNA Copy Number Calculator from the following website http://endmemo.com/bio/dnacopynum.php. To measure the *circNDUFB2* copy per cell, total RNA extracted from $1 \times 10^6$ A549 cells or H1299 cells were reversely transcribed into cDNAs, and aliquots of cDNA from 10,000 cells were further used for qRT-PCR. The copy number of *circNDUFB2* was quantitated from the standard curve.

**Northern blotting analysis**. Northern blotting was performed according to the manufacturer's protocol (DIG Northern Starter Kit, Roche). In brief, 10 μg of total RNA was loaded on a 1% agarose gel and transferred to a Hybond-N+ membrane (Amersham) and UV-crosslinked using the standard manufacturer's protocol. Membrane was then hybridized with a specific Dig-labeled probe. Northern blotting probe is listed in Supplementary Data 4.

**RNA pull-down assays**. For RNA pull-down assay, $1 \times 10^7$ cells were washed in ice-cold PBS, lysed in 500 μl co-IP buffer (Thermo Scientific) supplemented with a cocktail of proteinase inhibitors, phosphatase inhibitors, and RNase inhibitor (Invitrogen), and then incubated with 3 μg biotinylated DNA oligo probes against *circNDUFB2* backsplice junction region (sense) or corresponding complementary probes (antisense) for 2 h at room temperature. A total of 50 μl washed Streptavidin C1 magnetic beads (Invitrogen) were added to each binding reaction and further incubated for another hour at room temperature. The beads were washed briefly with co-IP buffer for five times. Finally, the retrieved proteins were used for mass spectrometry or western blot analysis. Probe sequences are listed in Supplementary Data 4.

**RNA immunoprecipitation**. RIP experiments were performed with a Magna RIP RNA-Binding Protein Immunoprecipitation Kit (Millipore, Billerica, MA, USA) according to the manufacturer's instructions. Co-precipitated RNA was detected by qRT-PCR.

**Gene-specific m6A qRT-PCR**. m6A modifications on *circNDUFB2* was determined using the Magna MeRIP m6A Kit (Millipore) according to the manufacturer's instructions. Briefly, 100 μg of total RNAs were sheared to about 100 nt in length by metal-ion-induced fragmentation, then purified and incubated with anti-m6A antibody-conjugated or mouse IgG-conjugated beads in 500 μl 1× immunoprecipitation buffer supplemented with RNase inhibitor at 4 °C overnight. Methylated RNA was immunoprecipitated with beads, eluted by competition with free m6A, and recovered with RNeasy Kit (Qiagen). One-tenth of the fragmented RNA was saved as input control, and further analyzed by qRT-PCR along with MeRIPed RNA. The related enrichment of m6A in each sample was calculated by normalizing to input.

**Co-immunoprecipitation**. To detect protein–protein interactions, cells were lysed in 500 μl co-IP buffer supplemented with a cocktail of proteinase inhibitors, phosphatase inhibitors, and RNase inhibitor. The lysates were centrifuged at 12,000$g$ for 30 min, and the supernatant was used for immunoprecipitation with beads, which were preincubated with the corresponding antibodies. After incubation at 4 °C overnight, beads were washed three times with co-IP buffer. SDS sample buffer was added to the beads and the immunoprecipitates were used for western blot analysis.

**Measurement of cytokines production**. Cell culture supernatants were collected and analyzed for cytokines production using ELISAs according to the manufacturer's instructions. ELISA kit for human CXCL10, CXCL11, CCL5, and IFNβ were from Arigobio Biolaboratories and mouse IFNβ was from R&D Systems.

**Analysis of tumor-infiltrating lymphocytes by flow cytometry**. Three weeks after implantation, tumors were dissected from the subcutaneous of mice, and treated with collagenase type IV (200 U/ml; Gibco) and DNase I (50 μg/ml; Sigma) for 20 min at 37 °C. Cells were passed through a 70-μm filter to remove clumps, diluted in medium, and a small aliquot taken directly for flow cytometry. Single-cell suspensions were blocked with anti-CD16/32 (1:100 dilution; Biolegend) for 20

min on ice and then incubated with appropriate antibodies (1:100 dilution) for 30 min on ice. Flow cytometry was performed on an LSRII (BD Biosciences) and data were analyzed using FlowJo software v 10.4.2 (FlowJo).

**Statistics**. Results are presented as mean ± s.d. Statistical analyses were performed using Prism software (GraphPad Software 8), and consisted of analysis of variance followed by Student's *t*-test when comparing two experimental groups. A probability of 0.05 or less was considered statistically significant.

**Reporting summary**. Further information on research design is available in the Nature Research Reporting Summary linked to this article.

## Data availability

Microarray data are accessible at the Gene Expression Omnibus (GEO) under accession number GSE158695. RNA-seq data are accessible at the GEO under accession number: GSE156607. Other data that support the findings of this study are available from the corresponding authors on reasonable request. Source data are provided with this paper.

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

## Acknowledgements

We thank Ling-Ling Chen for providing pZW1-FCS-circRNA plasmid. This work was supported by grants from National Natural Science Foundation of China (81772461, 81472175), National Key Research and Development Program of China (No. 2017YFC0907905, No. 2018YFC1313600), Shanghai Municipal Commission of Health and Family Planning (No.17411971100), and State Key Laboratory of Oncogenes and Related Gene (91-17-12).

## Author contributions

W.Q. supported and supervised the study. L.J. collected the NSCLC tissues and supervised the study. B.L., L.Z., and C.L. performed all the experimental validation assays and wrote the manuscript. C.W., H.W., H.J., and X.M. provided constructive comments and discussion. Z.C., C.Y., S.W., Q.Z., Y.Z., J.W., C.Y., and Y.L. analyzed data.

## Competing interests

The authors declare no competing interests.

## Additional information

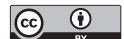

