## [Peer Review File · Nature Communications]

REVIEWER COMMENTS

Reviewer #1 (Lung cancer, therapy.) (Remarks to the Author):

This manuscript by Li et al includes a tremendous amount of data to make the case that circNDUFB2 inhibits growth and metastasis of NSCLC by enhancing interaction of TRIM25 with IGF2BPs with resultant ubiquitination. This data is stronger than the data for induction leading to an immune response based on recruitment of immune cells into the tumor microenvironment.

Concerns include:

- 1) In supplemental table two the cutoff between low and high should be stated somewhere. Similarly, a justification for the selected cutoff should be given. Also, note that over 2/3 of the patients were non-smokers, which is very unusual for lung cancer. Is there a reason for this? Is there any correlation with EGFR mutational status or was that data not available? Also, the staging information has boxes rather than numbers. Also, lymphatic invasion is generally noted on pathology from surgery. How would the authors have this data on advanced stage patients throughout. This information is of particular importance, as there were over 100 original circRNA changes. So, the strength of the findings are mainly based on the confirmation on these 52 patients.
- 2) The findings in figure 2 may reach significance, but they are not profound. The authors should discuss this.
- 3) In many places, the authors used two or more cell lines. However, there are a few places where they used only a single. It would be reassuring to see data replicated in multiple cell lines or at least have an explanation as to why only one cell line was used and why that was adequate for the specific experiment.
- 4) The models assessing immune response were questionable in terms of appropriateness to address the questions being asked. Also, some of the findings seemed random. Although enrichment of interferon signaling is potentially relevant, what was the framework in which they were evaluating expression of immune related genes?

Minor-

- 1) Although the English is generally excellent. There are a few areas that were presumably added after an English check where the language is awkward.
- 2) I don't believe that lung cancer is the most commonly diagnosed cancer.
- 3) More explanation of the METTL3/14 knockdown would be helpful.
- 4) Does figure 4f support their conclusions from it?

Reviewer #2 (circular RNA, non-coding RNA) (Remarks to the Author):

In this manuscript, Li et al. identified that circNDUFB2, a down-regulated circRNA in non-small cell lung cancer (NSCLC) tissues, inhibited growth and metastasis of NSCLC cells in vitro and in vivo. They also showed two aspects of mechanisms of action of circNDUFB2: On one hand, they found that circNDUFB2 promoted ubiquitination degradation of IGF2BPs by enhancing interaction of TRIM25 with IGF2BPs in an m6A modification-dependent manner. On the other hand, overexpression of circNDUFB2 triggered anti-tumor immune response in NSCLC cells by activating the RIG-I signaling cascade.

The work presented here is interesting because it provides a new mode of action of circular RNA in the context of NSCLC progression. While the amount of data presented are impressive, a number of issues need to be resolved and relevant controls should be added. Please see specific comments below.

Specific concerns:

1. The authors have shown that circNDUFB2 can directly interact with multiple abundant RNA binding proteins, including IGF2BPs and RIG-I in cells. How many copies of circNDUFB2 and these RBPs are expressed in cells? Can the stoichiometry of these molecules support the proposed models? If not,

how do the authors propose that this mechanism would operate?

2. Fig. 1h, 3c and 4b. How did the authors design the probe used in FISH experiment to show the localization of circNDUFB2? Of note, as the shared primary sequences between circNDUFB2 and its cognate mRNA, it's difficult to specifically target circNDUFB2 without off-target on its cognate mRNAs. How do they exclude the FISH signals are not circNDUFB2 cognate mRNAs, in particular, they both are localized in the cytoplasm?

3. Only one siRNA targeting circNDUFB2 BSJ was successfully used for their functional and mechanistic studies. It's not enough considering the potential off-target of siRNA. Additional siRNAs targeting circNDUFB2 BSJ are needed to confirm their results. Alternatively, other method-mediated circRNA KD can be done, such as ASO or Cas13-based KD.

4. For circNDUFB2 overexpression experiments throughout the manuscript (i.e. Fig. 2c, 2e, 2g, 2h as well as related assays in Fig. 5 and 6, see belowpoint #6), the authors should evaluate the amount of circNDUFB2, the pre-RNA and linear RNAs produced from the ectopic vectors by Northern Blot. This is important, as the authors must demonstrate the phenotypes that they overserved are not due to linear by-products from the circNDUFB2 ectopic vector.

5. Fig. 3f (circNDUFB2 OE), 3g (circNDUFB2 KD), why different cell lines were used in these assays

6. Fig. 5 and 6. The authors have suggested that circNDUFB2 could trigger anti-tumor immune response in NSCLC cells, as shown by circNDUFB2 overexpression experiments. Given that overexpression experiments can bring artifacts due to much high level of overexpressed circNDUFB2 and its linear by-products, it's necessary to confirm their results by siRNA-mediated circNDUFB2 KD.

7. Fig. 6g-i. The authors overexpressed the human circNDUFB2 in murine lung carcinoma cell (LLC1) to explore the immune response of circNDUFB2. Is circNDUFB2 conserved in mouse? If yes, does mouse circNdufb2 also play a role in cell growth and anti-tumor immune response? If not, what is the underlying reason for the phenotype of human circNDUFB2 in murine LLC1 cells?

8. Is the RIG-I-mediated, circNDUFB2-elicited immune response dependent on m6A modification of circNDUFB2? From the model shown Fig. 7, it seems that the RIG-I-mediated immune response requires circNDUFB2 m6A modification, but there is no any evidence to prove this at the moment in the MS.

9. Technical concerns: First, controls are missing in the manuscript. For example, for circRNA KD, control siRNAs with half-sequences replaced by scrambled sequences are recommended to exclude an influence on cognate linear RNAs. Second, in Fig. 3a, negative control such as β -ACTIN should be tested. Third, in Fig. 3b and 6b, both negative and positive controls should be added in RIP assays to show the specificity. Fourth, it will be more informative if quantification of WB panels in Fig. 3 and 4 could be provided. For example, in Fig. 4f, the authors claimed that protein levels of IGF2BPs were reduced upon overexpression of TRIM25 but not its truncation; however, there is weak difference on IGF2BPs expression after Flag-TRIM25 overexpression. Fifth, in Fig. 3a, were RNAs used in RNA pull-down assays circularized without contamination of cognate linear RNAs? This needs to be confirmed.

Point-by-Point Response to Referees' Comment

We are truly grateful to the reviewers for providing constructive and thoughtful comments, which have helped us to significantly improve this manuscript. We have addressed the specific points made by each of the reviewers and have incorporated their suggestions into the revised manuscript, as detailed in the point-by-point response below.

Reviewer #1 (Lung cancer, therapy.) (Remarks to the Author):

This manuscript by Li et al includes a tremendous amount of data to make the case that circNDUFB2 inhibits growth and metastasis of NCSLC by enhancing interaction of TRIM25 with IGF2BPs with resultant ubiquitination. This data is stronger than the data for induction leading to an immune response based on recruitment of immune cells into the tumor microenvironment.

Concerns include:

1) In supplemental table two the cutoff between low and high should be stated somewhere. Similarly, a justification for the selected cutoff should be given.

Response: Thank you very much for your careful concerns. Median expression level of a gene is a widely accepted cutoff when analyzing the correlation between its expression level and clinicopathologic features of tumor patients ^[1, 2]. As such, the current study used the median expression level of *circNDUFB2* (normalized by *GAPDH*) as the cutoff for high and low expressions. We identified *circNDUFB2* level >0.001157 as significantly higher expression, while < 0.001157 as lower expression. We have added this description into our revised supplementary information (Supplementary Table 2, page 14, line 25-26).

Also, note that over 2/3 of the patients were non-smokers, which is very unusual for lung cancer. Is there a reason for this?

Response: It is exactly as you mentioned, the association between lung cancer and smoking is generally not disputed ^[3]. We speculate that the following two reasons may result in over 2/3 (36/52) of the patients were non-smokers in this study. 1) In the current study, 3/4 (39/52) of the patients were LUAD. As we know, LUAD is not as strongly associated with smoking as LUSC, it is generally considered to be the dominant subtype in never-smokers with low tobacco exposure ^[4]. 2) There were 30 male patients and 22 female patients in our cohort. Among them, 15 male patients were non-smokers and 15 male patients were smokers. However, only one female patient was smoker. The majority of female patients (21 cases) were non-smokers may contribute to the phenomenon that most of the patients were non-smokers in our study.

Is there any correlation with EGFR mutational status or was that data not available?

Response: Thank you for your concern. Unfortunately, we can't get the information of EGFR mutational status for these patients in this study.

Also, the staging information has boxes rather than numbers.

Response: We are sorry for this mistake. We have corrected these errors in our revised manuscript (Supplementary Table 2, 3).

Also, lymphatic invasion is generally noted on pathology from surgery. How would the authors have this data on advanced stage patients throughout. This information is of particular importance, as there were over 100 original circRNA changes. So, the strength of the findings are mainly based on the confirmation on these 52 patients.

Response: Thank you for your comments. All patients in this study had performed preoperative examination such as CT and endobroncheal ultrasonography according to NCCN guidelines for NSCLC. After careful evaluation by our clinical multidiscipline team, they were able to receive surgery. There were 21 patients with stages III~IV (including: 18 patients with stage IIIA; 2 patients with stage IIIB; 1 patient with stage IV and this patient was identified to have pleural metastasis during surgery, which cannot be clearly detected by chest CT, and was finally classified as stage IV). The patients in this study had undergone surgery procedure, allowing us to assess lymphatic invasion status with their surgical samples.

References:

[¹] Cheng, Z. et al. circTP63 functions as a ceRNA to promote lung squamous cell carcinoma progression by upregulating FOXM1. *Nat. Commun.* 10, 3200 (2019).

[²] Jin, H. et al. Regulator of Calcineurin 1 Gene Isoform 4, Down-regulated in Hepatocellular Carcinoma, Prevents Proliferation, Migration, and Invasive Activity of Cancer Cells and Metastasis of Orthotopic Tumors by Inhibiting Nuclear Translocation of NFAT1. *Gastroenterology.* 153, 799-811 (2017).

[³] Hoffman, P.C. et al. Lung cancer. *Lancet.* 355, 479–85 (2000).

[⁴] Gridelli C. et al. Non-small-cell lung cancer. *Nat Rev Dis Primers.* 1, 15009 (2015).

2) The findings in figure 2 may reach significance, but they are not profound. The authors should discuss this.

Response: Thank you for your comments. We repeated these experiments of figure 2 several times, and all of the data in figure 2 showed the statistical significance. To address your concern, the other two siRNAs targeting the backsplice junction region were designed. si-*circNDUFB2#4* successfully silenced *circNDUFB2* and didn't alter the mRNA levels of *NDUFB2* in NSCLC cells. Consistently, knockdown of *circNDUFB2* promoted proliferation, migratory and invasion of NSCLC cells dramatically and significantly. We added these results into our revised manuscript (page 8, line 15-20; and page 9, line 2-4) and Figure S3h-j as well as their legends.

3) In many places, the authors used two or more cell lines. However, there are a few places where they used only a single. It would be reassuring to see data replicated in multiple cell lines or at least have an explanation as to why only one cell line was used and why that was adequate for the specific experiment.

Response: Thank you so much for your kind advice. According to your comments, we replicated our experiments in other NSCLC cell lines.

First, we further performed nuclear and cytoplasmic fractionation examination to confirm *circNDUFB2* was mainly localized in the cytoplasm of H1650 cells. We added the result into our revised Figure 1g and its legend.

Second, we further conducted tumor growth assays in H1299 cells. As result shown in Figure S3k, *circNDUFB2* overexpression could markedly inhibit tumorigenicity of H1299 cells. We added the result into our revised manuscript (page 9, line 4) and Figure S3k as well as its legend.

Third, we further conducted RNA pull down assay to confirm the association of *circNDUFB2* with TRIM25 in H1299 cells. Results showed that *circNDUFB2* could bind to TRIM25 in H1299 cells. We added the result into our revised manuscript (page 11, line 21) and Figure S5a as well as its legend.

Fourth, we further detected the expression levels of immune gene in H1299 cells with *circNDUFB2* overexpression. We observed that *circNDUFB2* overexpression significantly increased the levels of immune gene expression in H1299 cells. This result was added into our revised Figure 5d and its legend.

4) The models assessing immune response were questionable in terms of appropriateness to address the questions being asked. Also, some of the findings seemed random. Although enrichment of interferon signaling is potentially relevant, what was the framework in which they were evaluation expression of immune related genes?

Response: Thank you for your comments. Innate immune system provides the first line of defense against pathogen infection, which also influences pathways involved in cancer immunosurveillance ^[1]. Cytosolic RNA sensors include the RIG-I-like receptors (RLRs) family members that detect pathogen-derived RNA in the cytoplasm. Upon recognition of RNA, the RLRs activate a signaling cascade that culminates in production of type I interferons (IFNs) and proinflammatory cytokines as well as chemokines. Type I IFNs then induce a large array of antiviral genes through the activation of the Janus kinase (JAK)-signal transducer and activator of transcription (STAT) pathway ^[2, 3]. Recent studies showed exogenous circRNA stimulates innate immunity gene expression *in vitro* and functions as potent adjuvant to induce anti-tumor immunity *in vivo* ^[4, 5].

In the current study, we performed GO_BP and KEGG pathways enrichment analysis as well as GSEA for *circNDUFB2* target genes, results showed that signaling pathways of target genes significantly enriched in antiviral immunity (Figure 5a-c). Next, a panel of immune gene, which belongs to classical downstream targets of JAK-STAT pathway and is involved in antiviral immune response, was confirmed by qRT-PCR (Figure 5d, e). The level of cytokines induced by *circNDUFB2* was also detected by ELISAs (Figure 5f, g). These results suggest that *circNDUFB2* is involved in immune defense responses in NSCLC cells.

References:

- ^[1] Iurescia, S. et al. Targeting cytosolic nucleic acid-sensing pathways for cancer immunotherapies. *Front. Immunol.* 9, 711 (2018).
- ^[2] Wu, J. et al. Innate Immune Sensing and Signaling of Cytosolic Nucleic Acids. *Annu. Rev. Immunol.* 32, 461–488 (2014).
- ^[3] Platanias, LC. et al. Mechanisms of type-I- and type-II-interferon-mediated signalling. *Nat. Rev. Immunol.* 5:375–86 (2005).
- ^[4] Chen, Y. G. et al. Sensing Self and Foreign Circular RNAs by Intron Identity. *Mol. Cell* 67, 228–238 (2017).
- ^[5] Chen, Y. G. et al. N6-Methyladenosine Modification Controls Circular RNA Immunity. *Mol. Cell* 76, 96–109 (2019).

Minor-

- 1) Although the English is generally excellent. There are a few areas that were presumably added

after an English check where the language is awkward.

Response: Thank you for your comment. We corrected the errors, such as “have” was replaced by “has” (page 4, line 8), “lead” by “leads” (page 4, line 12) etc. in our revised manuscript.

2) I don't believe that lung cancer is the most commonly diagnosed cancer.

Response: Thank you very much for your comment. As reported by the Global Cancer Statistics 2018 ^[1], lung cancer is the most commonly diagnosed cancer (2,093,876 cases, 11.6% of the total cases) and the leading cause of cancer death (1,761,007 cases, 18.4% of the total cancer deaths) for both sexes combined. In addition, lung cancer is the most commonly diagnosed cancer and the leading cause of cancer death in males. Among females, breast cancer is the most commonly diagnosed cancer and the leading cause of cancer death ^[1]. According to your advice, we rephrased this sentence as “Lung cancer is one of the most commonly diagnosed cancer and the leading cause of cancer deaths in most countries” in the “Introduction” section of our revised manuscript (page 3, line 2-3).

Reference:

^[1] Bray, F. et al. Global cancer statistics 2018: GLOBOCAN estimates of incidence and mortality worldwide for 36 cancers in 185 countries. *CA. Cancer J. Clin.* 68, 394–424 (2018).

3) More explanation of the METTL3/14 knockdown would be helpful.

Response: Thank you for your kindly suggestion. N⁶-methyladenosine (m⁶A) modification is the most abundant and reversible internal modification of RNAs in eukaryotes ^[1, 2]. The m⁶A modification is installed by a multicomponent methyltransferase complex (MTC), which is composed of a core component *methyltransferase-like 3 (METTL3)* and *methyltransferase-like 14 (METTL14)* heterodimer ^[3]. In the current study, we found *circNDUFB2* contained m⁶A modification, and the level of m⁶A modification in *circNDUFB2* was significantly reduced upon *METTL3/14* knockdown (Figure 3e). Since IGF2BPs are known as m⁶A-binding proteins and the capability of IGF2BPs binding to its targets can be affected by the level of m⁶A modification in RNAs, we wondered if interactions between *circNDUFB2* and IGF2BPs were regulated via an m⁶A-dependent manner. We performed RNA pull down in A549 cells, results showed the interactions between *circNDUFB2* and IGF2BPs were remarkably impaired by *METTL3/14* knockdown (Figure 3d). Furthermore, *circNDUFB2* functions as a scaffold to enhance interaction of TRIM25 with IGF2BPs, subsequently facilitates ubiquitination degradation of IGF2BPs. Knockdown of *METTL3/14* reduced ubiquitination efficiency of TRIM25 for IGF2BPs (Figure S5o-q). These results reveal that *circNDUFB2* interacts with IGF2BPs and facilitates the TRIM25-mediated ubiquitination of IGF2BPs in an m⁶A-dependent manner. According to your suggestion, we added the description in our revised manuscript (page 10, line 16-19).

References:

^[1] Fu, Y., Dominissini, D., Rechavi, G. & He, C. Gene expression regulation mediated through reversible m⁶A RNA methylation. *Nat. Rev. Genet.* 15, 293–306 (2014).

^[2] Zhou, C. et al. Genome-Wide Maps of m⁶A circRNAs Identify Widespread and Cell-Type-Specific Methylation Patterns that Are Distinct from mRNAs. *Cell Rep.* 20, 2262–2276 (2017).

^[3] Huang, H. et al. m⁶A Modification in Coding and Non-coding RNAs: Roles and Therapeutic Implications in Cancer. *Cancer cell.* 37.270-288 (2020).

4) Does figure 4f support their conclusions from it?

Response: Thank you for your question. TRIM25 is a RNA-binding protein and belongs to the Tripartite Motif (TRIM) family of E3 ubiquitin ligases, which catalyses the addition of polyubiquitin chains to its substrates for degradation ^[1, 2]. We found protein levels of IGF2BPs were dramatically increased in TRIM25 knockdown and reduced in TRIM25 overexpression, respectively (Figure 4e, f). Meanwhile, ubiquitination levels of IGF2BPs were dramatically increased in A549 cells with TRIM25 overexpression (Figure S5h-j). These data suggest that TRIM25 promotes proteasome degradation of IGF2BPs. Since TRIM25 uses RNA as a scaffold for efficient ubiquitination of its targets and the RNA-binding activity of TRIM25 is essential for its ubiquitin-ligase activity towards the substrate ^[3]. We then constructed a RNA-binding domain (RBD) deletion mutant of TRIM25 with FLAG-tagged (termed TRIM25ΔRBD), results showed that TRIM25ΔRBD didn't affect protein levels of IGF2BPs (Figure 4f), and no obvious effect on ubiquitination level of IGF2BPs was detected (Figure S5h-j). Altogether, these results suggest that TRIM25 is the E3 ubiquitin ligase that mediates the ubiquitination of IGF2BPs, and the intact RBD of TRIM25 is necessary for efficient ubiquitination and degradation of IGF2BPs.

References:

^[1] Urano, T. et al. Efp targets 14-3-3σ for proteolysis and promotes breast tumour growth. *Nature* 417, 871–876 (2002).

^[2] Lee, J. M. et al. The E3 ubiquitin ligase TRIM25 regulates adipocyte differentiation via proteasome-mediated degradation of PPARγ. *Exp. Mol. Med.* 50, 1–11 (2018).

^[3] Choudhury, N. R. et al. RNA-binding activity of TRIM25 is mediated by its PRY/SPRY domain and is required for ubiquitination. *BMC Biol.* 15, 1–20 (2017).

Reviewer #2 (circular RNA, non-coding RNA) (Remarks to the Author):

In this manuscript, Li et al. identified that circNDUFB2, a down-regulated circRNA in non-small cell lung cancer (NSCLC) tissues, inhibited growth and metastasis of NSCLC cells in vitro and in vivo. They also showed two aspects of mechanisms of action of circNDUFB2: On one hand, they found that circNDUFB2 promoted ubiquitination degradation of IGF2BPs by enhancing interaction of TRIM25 with IGF2BPs in an m6A modification-dependent manner. On the other hand, overexpression of circNDUFB2 triggered anti-tumor immune response in NSCLC cells by activating the RIG-I signaling cascade.

The work presented here is interesting because it provides a new mode of action of circular RNA in the context of NSCLC progression. While the amount of data presented are impressive, a number of issues need to be resolved and relevant controls should be added. Please see specific comments below.

Specific concerns:

1. The authors have shown that circNDUFB2 can directly interact with multiple abundant RNA binding proteins, including IGF2BPs and RIG-I in cells. How many copies of circNDUFB2 and

these RBPs are expressed in cells? Can the stoichiometry of these molecules support the proposed models? If not, how do the authors propose that this mechanism would operate?

Response: Thank you for your good question. We calculated the copy numbers of IGF2BPs, RIG-I and *circNDUFB2* in NSCLC cells by absolute quantification (the standard curves of RIG-I and IGF2BPs see the following Figures a-d). Measurement of protein molecular number of IGF2BPs in NSCLC cells revealed 2221 molecules and 1842 molecules per A549 cell and H1299 cell, respectively. Examination of protein molecular number of RIG-I showed per A549 cell and H1299 cell contained 50 molecules and 286 molecules, respectively. *circNDUFB2* calculation revealed that per A549 cell contained 151 molecules and per H1299 cell contained 206 molecules of *circNDUFB2*. Considered that *circNDUFB2* has a long half-life contribute it to work longer as functional molecule and with two IGF2BPs binding motifs of per *circNDUFB2*, *circNDUFB2* has the capacity to bind a considerable fraction of IGF2BPs and RIG-I thus promote the degradation of IGF2BPs and activation of anti-tumor immunity. We added these results into our revised manuscript (page 18, line 4-9) and Figure S7l-m as well as their legends.

(a-d) Linear relationship between the mass of protein and its signals intensity by western blot analysis.

2. Fig. 1h, 3c and 4b. How did the authors design the probe used in FISH experiment to show the localization of *circNDUFB2*? Of note, as the shared primary sequences between *circNDUFB2* and its cognate mRNA, it's difficult to specifically target *circNDUFB2* without off-target on its cognate mRNAs. How do they exclude the FISH signals are not *circNDUFB2* cognate mRNAs, in particularly, they both are localized in the cytoplasm?

Response: Thank you for your good comments. Probe used in FISH experiment was designed to against *circNDUFB2* backsplice junction region (probe sequence was listed in Supplementary Table 7). In order to exclude the FISH signals were not *circNDUFB2* cognate mRNAs, we performed FISH experiment in A549 cells with or without RNase R treatment. Results showed that FISH signals didn't weaken in the cells with RNase R treatment. However, the FISH signals were barely detectable in the cells with *circNDUFB2* knockdown. These results indicated that the FISH signals were *circNDUFB2* rather than its cognate mRNAs. We revised the previous Figure 1h and its legend.

3. Only one siRNA targeting *circNDUFB2* BSJ was successfully used for their functional and mechanistic studies. It's not enough considering the potential off-target of siRNA. Additional siRNAs targeting *circNDUFB2* BSJ are needed to confirm their results. Alternatively, other method-mediated *circRNA* KD can be done, such as ASO or Cas13-based KD.

Response: Thank you so much for your good advice. According to your suggestion, we designed the other two siRNAs (si-*circNDUFB2*#3 and si-*circNDUFB2*#4) to further exclude the off-target effect and confirm the function of *circNDUFB2*. Meanwhile, take into consideration of your

comments in point #9, a control siRNA (si-NC#2) with half-sequences replaced by scrambled sequences was designed. Results showed that si-*circNDUFB2*#4 successfully silenced *circNDUFB2* and didn't alter the level of *NDUFB2* mRNA. Functionally, *circNDUFB2* knockdown by si-*circNDUFB2*#4 significantly promoted the proliferation, migration and invasion of H1650 and H460 cells *in vitro*, which is consistent with the effect of si-*circNDUFB2*#1. Besides, knockdown of *circNDUFB2* by si-*circNDUFB2*#4 had no effect on IGF2BPs mRNA, but protein levels of IGF2BPs in NSCLC cells were dramatically increased upon *circNDUFB2* knockdown. We added these results into our revised Figure S3h-j, Figure S4k-l and their legends.

4. For *circNDUFB2* overexpression experiments throughout the manuscript (i.e. Fig. 2c, 2e, 2g, 2h as well as related assays in Fig. 5 and 6, see below point #6), the authors should evaluate the amount of *circNDUFB2*, the pre-RNA and linear RNAs produced from the ectopic vectors by Northern Blot. This is important, as the authors must demonstrate the phenotypes that they observed are not due to linear by-products from the *circNDUFB2* ectopic vector.

Response: Thank you so much for your good comment. The ectopic vector used in our overexpression experiments is pZW-FCS-*circRNA* (a gift from Ling-Ling Chen, Shanghai Institutes for Biological Sciences). The *circPOLR2A* circularization efficiency of this ectopic vector was approximately 47% in HeLa cells (showed in Figure 3D and Figure 4D of Zhang's paper ^[1]), and approximately 78% in NIH 3T3 cells (showed in Figure S3C of Zhang's paper ^[1]). According to your advice, we conducted northern blotting analysis to evaluate the amount of *circNDUFB2* and linear RNAs produced from pZW1-FCS-*circNDUFB2* plasmid. Results showed that exon circularization efficiency of the pZW1-FCS-*circNDUFB2* plasmid is 56% and 53% in A549 cells and H1299 cells, respectively. We added these results into our revised manuscript (page 8, line 12-14) and Figure S3g as well as its legend.

To further confirm that the phenotypes in *circNDUFB2* overexpression experiments were caused by *circNDUFB2* instead of linear by-products from the pZW1-FCS-*circNDUFB2*, we constructed a linear-*NDUFB2* expression vector (pZW1-FCS-linear*NDUFB2*), which deleted the upstream complementary sequences in pZW1-FCS-*circNDUFB2* plasmid. We detected pZW1-FCS-*circNDUFB2* produced both *circNDUFB2* and linear-*NDUFB2*, whereas the pZW1-FCS-linear*NDUFB2* only produced linear-*NDUFB2* (see the following Figure a). Subsequently, *in vitro* studies showed that pZW1-FCS-linear*NDUFB2* didn't affect proliferation, migration and invasion of NSCLC cells (see the following Figure b-c). In addition, pZW1-FCS-linear*NDUFB2* didn't provoke cellular immune response in NSCLC cells (see the following Figure d). These results indicated that overexpression of *circNDUFB2* rather than its linear RNA fragment with the identical sequence led to activate anti-tumor immunity and inhibit NSCLC progression.

(a) Top: A sketch map for pZW1-FCS-linearNDUFB2 plasmid construction. Bottom: qRT-PCR showed pZW1-FCS-circNDUFB2 plasmid produced both *circNDUFB2* and linear-*NDUFB2*, whereas pZW1-FCS-linearNDUFB2 plasmid only produced linear-*NDUFB2*. (b) Cell proliferation assays for NSCLC cells with linear-*NDUFB2* overexpression. (c) Migration and invasion assays for NSCLC cells with linear-*NDUFB2* overexpression. (d) Expression levels of indicated mRNA in NSCLC cells with *circNDUFB2* or linear-*NDUFB2* overexpression. Error bars indicate s.d. ns means no significant difference. *** $P<0.001$.

Reference:

[1] Zhang, X. O. et al. Complementary Sequence-Mediated Exon Circularization. *Cell*. 159, 134-147 (2014).

5. Fig. 3f (circNDUFB2 OE), 3g (circNDUFB2 KD), why different cell lines were used in these assays?

Response: Thank you for your question. In general, cell lines with relatively high endogenous expression levels of a gene were selected for knockdown experiments, while cell lines with relatively low endogenous expression levels of a gene were selected for overexpression experiments [1-3]. Data in Figure 3f and Figure 3g were used to detect the influence of *circNDUFB2* on IGF2BPs protein level. As shown in Figure S3a, the content of endogenous *circNDUFB2* in A549 and H1299 cells were relatively low, so these two cell lines were selected for the *circNDUFB2* overexpression assays. H1650 and H460 cells contained relatively high level of endogenous *circNDUFB2*, so these two cell lines were used for *circNDUFB2* knockdown assays.

References:

[1] Cheng, Z. et al. circTP63 functions as a ceRNA to promote lung squamous cell carcinoma

progression by upregulating FOXM1. Nat. Commun. 10, 3200 (2019).

[2] Jin, H. et al. Regulator of Calcineurin 1 Gene Isoform 4, Down-regulated in Hepatocellular Carcinoma, Prevents Proliferation, Migration, and Invasive Activity of Cancer Cells and Metastasis of Orthotopic Tumors by Inhibiting Nuclear Translocation of NFAT1. Gastroenterology. 153, 799-811 (2017).

[3] Yu, J. et al. Circular RNA cSMARCA5 inhibits growth and metastasis in hepatocellular carcinoma. J. Hepatol. 68, 1214–1227 (2018).

6. Fig. 5 and 6. The authors have suggested that circNDUFB2 could trigger anti-tumor immune response in NSCLC cells, as shown by circNDUFB2 overexpression experiments. Given that overexpression experiments can bring artifacts due to much high level of overexpressed circNDUFB2 and its linear by-products, it's necessary to confirm their results by siRNA-mediated circNDUFB2 KD.

Response: Thank you so much for your valuable advice. As shown in Figure 5a-d and Figure 5f, *circNDUFB2* overexpression promoted expression of anti-tumor immune related genes and increased the levels of cytokines in NSCLC cells. In addition, we found *circNDUFB2* knockdown reduced the levels of these cytokines in H1650 cells (Figure 5g). According to your advice, we knockdown *circNDUFB2* in H1650 cells, results showed that knockdown of *circNDUFB2* by si-*circNDUFB2*#1 and si-*circNDUFB2*#4 significantly reduced the expression of anti-tumor immune related genes. We added the result into our revised manuscript (page 14, line 19-20) and Figure 5e as well as its legend.

7. Fig. 6g-i. The authors overexpressed the human circNDUFB2 in murine lung carcinoma cell (LLC1) to explore the immune response of circNDUFB2. Is circNDUFB2 conserved in mouse? If yes, does mouse circNdufb2 also play a role in cell growth and anti-tumor immune response? If not, what is the underlying reason for the phenotype of human circNDUFB2 in murine LLC1 cells?

Response: Thank you for your good question. *circNDUFB2* is conserved in mouse. We analyzed the circRNAs derived from *Ndufb2* in circBank (<http://www.circbank.cn>) and found a conserved *circNdufb2* with a length of 245 nt in mice (see the following Table 1). Similar to the human *circNDUFB2*, mouse *circNdufb2* is derived from the second and third exons of *Ndufb2* gene, and the sequence similarity between *circNDUFB2* and *circNdufb2* is 80.7% (201/249) (see the following Figure a). The backsplice junction site of *circNdufb2* was amplified using divergent primers and confirmed by Sanger sequencing (see the following Figure b).

As shown in Figure 6g-i, *circNDUFB2* overexpression could markedly inhibit tumorigenicity of LLC1 cells and promote infiltration of CD8⁺ T cells and DCs into tumor microenvironment. According to your suggestion, we explored whether *circNdufb2* also played a role in cell growth and anti-tumor immune response in LLC1 cells. For *circNdufb2* knockdown, two siRNAs specifically targeting the backsplice junction region were designed, both si-*circNdufb2*#1 and si-*circNdufb2*#2 successfully silenced *circNdufb2* and didn't alter the mRNA level of *Ndufb2* in LLC1 cells. We next constructed *circNdufb2* overexpression plasmid and confirmed *circNdufb2* was overexpressed efficiently (see the following Figure c). Consistent with *circNDUFB2*, *circNdufb2* knockdown significantly promoted proliferation and reduced the levels of anti-tumor immune related genes in LLC1 cells, while *circNdufb2* overexpression reversed

these effects (see the following Figure d, e).

Table 1. Analysis for the circRNAs derived from *Ndufb2* in circBank

CircBase_ID	Position	Length (nt)	Gene Symbol	Conserved mouse circRNA
hsa_circ_0007518	chr7: 140402665-140404763	249	NDUFB2	chr6: 39596450_39598361
hsa_circ_0082730	chr7: 140396480-140404763	411	NDUFB2	No annotation
hsa_circ_0082731	chr7: 140396480-140406446	495	NDUFB2	No annotation
hsa_circ_0082732	chr7: 140397874-140404763	5040	NDUFB2	No annotation
hsa_circ_0082733	chr7: 140402665-140402810	145	NDUFB2	No annotation
hsa_circ_0082734	chr7: 140402665-140406446	333	NDUFB2	No annotation
hsa_circ_0082735	chr7: 140404659-140404763	104	NDUFB2	No annotation
hsa_circ_0082736	chr7: 140404659-140406446	188	NDUFB2	No annotation

(a) Sequence similarity analysis for *circNDUFB2* and *circNdufb2*. (b) The backsplice junction site of *circNdufb2* was identified by Sanger sequencing. (c) Expression levels of *circNdufb2* and *Ndufb2* in LLC1 cells with *circNdufb2* knockdown or overexpression. (d) Cell proliferation assays for LLC1 cells with *circNdufb2* knockdown or overexpression. (e) Expression levels of indicated mRNA in LLC1 cells with *circNdufb2* knockdown or overexpression. Error bars indicate s.d. ns means no significant difference. * $P < 0.05$, ** $P < 0.01$, *** $P < 0.001$.

8. Is the RIG-I-mediated, circNDUFB2-elicited immune response dependent on m⁶A modification of circNDUFB2? From the model shown Fig. 7, it seems that the RIG-I-mediated immune response requires circNDUFB2 m⁶A modification, but there is no any evidence to prove this at the moment in the MS.

Response: Thanks a lot for your good comments. It has been reported that m⁶A is deposited on native RNA transcripts during transcription and backsplicing of circRNAs-forming exons largely occurred post-transcriptionally [1-3]. Here, we revealed that the level of m⁶A modification in *circNDUFB2* impacted the strength of *circNDUFB2* binding to IGF2BPs (Figure 3d), and then affected ubiquitin ligase activity of TRIM25 for IGF2BPs (Figure S5o-q). In our first submission, we didn't observe whether the RIG-I-mediated immune response is regulated by the m⁶A modification in *circNDUFB2*. According to your advice, we explored whether RIG-I binds

circNDUFB2 at m⁶A modification site. We performed RNA pull down assay in NSCLC cells with *circNDUFB2* or *circNDUFB2* mutant (Figure S4g) overexpression. We noticed that mutation of m⁶A modification site in *circNDUFB2* didn't affect its interaction with RIG-I (see the following Figure a). Next, the *circNDUFB2* mutant was transfected into NSCLC cells to detect expression levels of anti-tumor immune related genes. Results showed that no significantly difference expression of anti-tumor immune related genes was induced between *circNDUFB2* and *circNDUFB2* mutant (see the following Figure b). These data suggest that RIG-I doesn't bind *circNDUFB2* at m⁶A modification site, and *circNDUFB2*-elicited immune response is not dependent on m⁶A modification in *circNDUFB2*.

(a) RNA pull down assay was performed using biotinylated sense probe for *circNDUFB2* in A549 cells with *circNDUFB2* or *circNDUFB2-MUT* overexpression. (b) Expression levels of indicated mRNAs in NSCLC cells with *circNDUFB2* or *circNDUFB2* mutant overexpression. Error bars indicate s.d. ns means no significant difference.

References:

- [1] Huang, H. et al. m⁶A Modification in Coding and Non-coding RNAs: Roles and Therapeutic Implications in Cancer. *Cancer cell*. 37.270-288 (2020).
- [2] Li, X. et al. The Biogenesis, Functions, and Challenges of Circular RNAs. *Mol. Cell* 71, 428–442 (2018).
- [3] Kristensen, L. S. et al. The biogenesis, biology and characterization of circular RNAs. *Nat. Rev. Genet.* 20, 675–691 (2019).

9. Technical concerns: First, controls are missing in the manuscript. For example, for circRNA KD, control siRNAs with half-sequences replaced by scrambled sequences are recommended to exclude an influence on cognate linear RNAs. Second, in Fig. 3a, negative control such as β -ACTIN should be tested. Third, in Fig. 3b and 6b, both negative and positive controls should be added in RIP assays to show the specificity. Fourth, it will be more informative if quantification of WB panels in Fig. 3 and 4 could be provided. For example, in Fig. 4f, the authors claimed that

protein levels of IGF2BPs were reduced upon overexpression of TRIM25 but not its truncation; however, there is weak difference on IGF2BPs expression after Flag-TRIM25 overexpression. Fifth, in Fig. 3a, were RNAs used in RNA pull-down assays circularized without contamination of cognate linear RNAs? This needs to be confirmed.

Response: Thank you so much for your valuable advice.

1) According to your comments, a control siRNA with half-sequences replaced by scrambled sequences (siNC#2) was designed and used for function experiments of *circNDUFB2*. We added these results into Figure S3h-j and their figure legends in revised submission.

2) Thanks, we added β -actin as a negative control. We added this result into Figure 3a and its legend.

3) For Fig. 3b, we added *IGF2* as a positive control and *TINCR* as a negative control (showed in Figure 4C of Hosono's paper ^[1]). For Fig. 6b, we added *POLR2A* as a positive control and *circSMARCA5* as a negative control (showed in Figure S6C of Liu's paper ^[2]).

4) According to your advice, western blot signals in Fig. 3 and 4 were quantified using the ImageJ program.

5) As descriptions in "Methods" section, for RNA pull down assays we designed biotinylated DNA oligo probe against *circNDUFB2* backsplice junction region (sense probe) and incubated this probe with endogenous *circNDUFB2* in cell lysates. Next, we performed qRT-PCR analysis and confirmed that sense probe for *circNDUFB2* could enrich *circNDUFB2* efficiently rather than *NDUFB2* mRNA (Figure S4b), these results could exclude contamination of cognate linear RNAs.

References:

^[1] Hosono, Y. et al. Oncogenic Role of THOR, a Conserved Cancer/Testis Long Non-coding RNA. *Cell*. 171, 1559-1572 (2017).

^[2] Liu, C. X. et al. Structure and Degradation of Circular RNAs Regulate PKR Activation in Innate Immunity. *Cell*. 177, 865–880 (2019).

Finally, we would like to thank reviewers again for your help to our manuscript.

REVIEWERS' COMMENTS

Reviewer #1 (Remarks to the Author):

I would like to thank the authors for both their excellent additions to the manuscript as well as making it very easy to see what has been changed. The remaining issues that I have at this point are minor.

1) I think that in the discussion, the authors should specifically acknowledge two things about the patients evaluated: 1) The smoking status is quite different from what is typically seen in the disease, and two, 2) The correlation between EGFR mutational status and their findings is not known. It is hard to know whether the findings should be considered to be associated with NSCLC, EGFR mutant NSCLC or non-smoking NSCLC.

2) A character is lost on line 586 of the revised manuscript (a symbol appears to have become a box).

Reviewer #2 (Remarks to the Author):

The revised manuscript is improved considerably, with the addition of a large amount of new data. While most issues have been appropriately addressed, some original questions remain to be solved prior to the publication of this manuscript.

1. Previous Question 1: The authors have quantified the copy numbers of circNDUFB2, IGF2BPs and RIG-I in cells. It's unexpected that only 50-1000 copies of these RBPs per cell, especially considering that IGF2BPs are known as very abundant proteins in cancer cell lines. In fact, >100,000 copies of IGF2BPs per HeLa cell were estimated by another group (PMID: 26496610). How do you explain this discrepancy? The authors should check their quantification results carefully, and discuss the stoichiometry of these molecules in the manuscript clearly. At the moment, the discussion on this point on page 18, lines 343-348 is superficial.

2. Previous Question 4: Inclusion of the linear-NDUFB2 expression vector (pZW1-FCS-linearNDUFB2) has addressed our concern that "the phenotypes that they observed are due to linear by-products from the circNDUFB2 ectopic vector". Therefore, the related results in the rebuttal are worthwhile to be included in the manuscript to further confirm their conclusions.

3. Previous Question 8: In the rebuttal letter, the authors have shown that the RIG-I-mediated, circNDUFB2-elicited immune response is independent of circNDUFB2 m6A modification. To avoid misunderstanding, please include these data in the manuscript. Of note, transcripts are only partially m6A modified considering the m6A modification efficiency. It should be the case for circNDUFB2 as well. This point should be also clarified in their model (Fig. 7).

Point-by-Point Response to Referees' Comment

We thank the referees for their insightful and constructive comments. We have addressed the specific points made by each of the reviewers and have incorporated their suggestions into the revised manuscript, as detailed in the point-by-point response below.

Reviewer #1 (Remarks to the Author):

I would like to thank the authors for both their excellent additions to the manuscript as well as making it very easy to see what has been changed. The remaining issues that I have at this point are minor.

1) I think that in the discussion, the authors should specifically acknowledge two things about the patients evaluated: 1) The smoking status is quite different from what is typically seen in the disease, and two, 2) The correlation between EGFR mutational status and their findings is not known. It is hard to know whether the findings should be considered to be associated with NSCLC, EGFR mutant NSCLC or non-smoking NSCLC.

Response: Thank you for your advice. According to your suggestion, we discuss the correlation between *circNDUFB2* level and smoking status in NSCLC. We also acknowledge the relevance between EGFR mutational status and *circNDUFB2* is unknown and needs to be studied in the future. We added the description in our revised manuscript (page 20, line 12-22 and page 21, line 1-2).

2) A character is lost on line 568 of the revised manuscript (a symbol appears to have become a box).

Response: We are sorry for this mistake. We have corrected.

Reviewer #2 (Remarks to the Author):

The revised manuscript is improved considerably, with the addition of a large amount of new data. While most issues have been appropriately addressed, some original questions remain to be solved prior to the publication of this manuscript.

1. Previous Question 1: The authors have quantified the copy numbers of *circNDUFB2*, IGF2BPs and RIG-I in cells. It's unexpected that only 50-1000 copies of these RBPs per cell, especially considering that IGF2BPs are known as very abundant proteins in cancer cell lines. In fact, >100,000 copies of IGF2BPs per HeLa cell were estimated by another group (PMID: 26496610). How do you explain this discrepancy? The authors should check their quantification results carefully, and discuss the stoichiometry of these molecules in the manuscript clearly. At the moment, the discussion on this point on page 18, lines 343-348 is superficial.

Response: Thank you so much for your kind advice. In order to check whether our methodology of protein molecular number absolute quantification is reliable, we calculated the copy number of IGF2BP1 and IGF2BP3 in HeLa cell. Results showed that 103720 copies of IGF2BP1 and 642722 copies of IGF2BP3 per HeLa cell (see the following Figure), these are similar with previous research results that each HeLa cell contained 111246 copies of IGF2BP1 and 913315 copies of IGF2BP3, respectively (Table S3 of Hein's paper ^[1]). These results show that our method of

protein absolute quantification is reliable and the results of RBPs copy number in A549 and H1299 cells should be credible.

As you mentioned, IGF2BPs are abundant in several types of aggressive cancers and many kinds of cancer cell lines. However, the abundance of IGF2BPs varies greatly among cancer cell lines [2]. For example, IGF2BP1/2/3 were expressed at high levels in PANC-1 (pancreas adenocarcinoma), HepG2 (hepatocellular carcinoma) and ES-2 (clear cell ovarian carcinoma) cells but did not detected in MCF7 (mammary adenocarcinoma) cell [2]. The abundance in the three proteins of IGF2BPs was also varies greatly in cells. For example, HeLa cell contained abundant IGF2BP1 and IGF2BP3, but IGF2BP2 was hardly expressed [2]. Whereas U-2OS (osteosarcoma) cell contained abundant IGF2BP2, but IGF2BP1 and IGF2BP3 were hardly detectable [2]. The discrepancy of RBPs copies number in our study may be attributable to heterogeneity in different cancer cell lines.

According to your advice, we added a description and meaningful discussion about the molecules stoichiometry in our revised manuscript (page 19, line 4-7 and line 10-14).

(a) Western blot analysis for purified recombinant IGF2BP1 protein and cell lysate in 50000 HeLa cells. (b) Western blot analysis for purified recombinant IGF2BP3 protein and cell lysate in 25000 HeLa cells. (c) Linear relationship between the mass of protein and its signals intensity by western blot analysis. (d) Quantification summarized in table.

Reference:

[1] Marco Y Hein. et al. A human interactome in three quantitative dimensions organized by stoichiometries and abundances. *Cell*. 163, 712-723 (2015).

[2] Bell, J. L. et al. Insulin-like growth factor 2 mRNA-binding proteins (IGF2BPs): Post-transcriptional drivers of cancer progression? *Cell. Mol. Life Sci*. 70, 2657–2675 (2013).

2. Previous Question 4: Inclusion of the linear-NDUFB2 expression vector (pZW1-FCS-linearNDUFB2) has addressed our concern that “the phenotypes that they observed are due to linear by-products from the circNDUFB2 ectopic vector”. Therefore, the related results in the rebuttal are worthwhile to be included in the manuscript to further confirm their conclusions.

Response: Thank you for your advice. According to your comments, we added these results into our revised manuscript (page 9, line 9-17 and page 15, line 9-12), Supplementary Figure 4, 8a as well as their legends.

3. Previous Question 8: In the rebuttal letter, the authors have shown that the RIG-I-mediated, circNDUFB2-elicited immune response is independent of circNDUFB2 m6A modification. To avoid misunderstanding, please include these data in the manuscript. Of note, transcripts are only partially m6A modified considering the m6A modification efficiency. It should be the case for circNDUFB2 as well. This point should be also clarified in their model (Fig. 7).

Response: Thank you so much for your valuable advice. According to your comments, we added these results into our revised manuscript (page 16, line 20-22 and page 17, line 1-7) and Supplementary Figure 8e-f as well as their legends. Moreover, considering that transcripts are commonly only partially m⁶A modified, our model has been revised to only part of *circNDUFB2* with m⁶A modification (Fig. 7).

Finally, we would like to express our great appreciation to reviewers again for your help to our manuscript.